# Eugenol mimics exercise to promote skeletal muscle fiber remodeling and myokine IL-15 expression by activating TRPV1 channel

**Tengteng Huang[†], Xiaoling Chen[†], Jun He, Ping Zheng, Yuheng Luo, Aimin Wu, Hui Yan, Bing Yu, Daiwen Chen, Zhiqing Huang***

Key Laboratory for Animal Disease-Resistance Nutrition of China Ministry of Education, Institute of Animal Nutrition, Sichuan Agricultural University, Chengdu, China

**\*For correspondence:**
zqhuang@sicau.edu.cn

[†]These authors contributed equally to this work

**Competing interest:** The authors declare that no competing interests exist.

**Abstract** Metabolic disorders are highly prevalent in modern society. Exercise mimetics are defined as pharmacological compounds that can produce the beneficial effects of fitness. Recently, there has been increased interest in the role of eugenol and transient receptor potential vanilloid 1 (TRPV1) in improving metabolic health. The aim of this study was to investigate whether eugenol acts as an exercise mimetic by activating TRPV1. Here, we showed that eugenol improved endurance capacity, caused the conversion of fast-to-slow muscle fibers, and promoted white fat browning and lipolysis in mice. Mechanistically, eugenol promoted muscle fiber-type transformation by activating TRPV1-mediated CaN signaling pathway. Subsequently, we identified IL-15 as a myokine that is regulated by the CaN/nuclear factor of activated T cells cytoplasmic 1 (NFATc1) signaling pathway. Moreover, we found that TRPV1-mediated CaN/NFATc1 signaling, activated by eugenol, controlled IL-15 levels in C2C12 myotubes. Our results suggest that eugenol may act as an exercise mimetic to improve metabolic health via activating the TRPV1-mediated CaN signaling pathway.

## eLife assessment

This **useful** paper addresses a novel exercise mimetic agent on muscle exercise and performance. While the data provided are interesting, the evidence is **incomplete**, as much of it is correlative.

## Introduction

Due to the lack of regular exercise among many individuals (*Carlson et al., 2010*), metabolic diseases are highly prevalent in modern society (*Hoehn et al., 2010*). To address this issue, pharmacological interventions have been considered as potential treatments. Exercise mimetics are pharmacological compounds that produce fitness benefits (*Fan and Evans, 2017*). Although some synthetic exercise mimetics such as AICAR (AMPK activator), GW501516 (PPARδ activator), SRT1720 (SIRT1 activator), and GSK4716 (ERRγ activator) have been used to improve fitness (*Feige et al., 2008*; *Narkar et al., 2008*; *Rangwala et al., 2010*), the safety and health of each drug must be considered. In recent years, plant extracts such as resveratrol, lycium barbarum, and epicatechin have demonstrated potential as a new class of low-toxin exercise mimetics (*Meng et al., 2020*; *Nogueira et al., 2011*; *Wen et al., 2020*). Therefore, it is of great significance to discover natural medicinal and edible plants that have mimetic effects of exercise.

Based on their metabolism and contractile properties, muscle fibers are commonly divided into slow oxidative fibers (muscle fibers that express myosin heavy chain (MyHC) I) and fast glycolytic/oxidative fibers (muscle fibers that express MyHC IIa, MyHC IIx, or MyHC IIb) (*Schiaffino and Reggiani, 2011*). Different muscle fibers exhibit a high degree of plasticity and are regulated by exercise (*Egan and Zierath, 2013*). The increased proportion of slow muscle fibers contributes to the enhancement of mitochondrial biogenesis and lipid metabolism (*Carlson et al., 2010*). Therefore, one of the benefits of exercise is to treat metabolic disorders by promoting slow muscle fiber remodeling (*Duan et al., 2017*). In addition, it has been increasingly recognized that myokines released by skeletal muscle play an essential role in mediating the benefits of exercise. Myokines are defined as cytokines or peptides that are released from skeletal muscle cells, which exert autocrine, paracrine, or endocrine effects (*Pedersen and Febbraio, 2012*). As a medium of crosstalk between muscles and other organs, myokines exert systemic regulation of exercise by acting on the muscle, fat, liver, pancreas, and other organs, effectively improving insulin resistance, obesity, and metabolic disorders of type 2 diabetes (*Severinsen and Pedersen, 2020*). Therefore, another potential effect of exercise is to induce the release of myokines (*Fan and Evans, 2017*). Although many previous studies have discovered plant extracts as exercise mimetics by promoting slow oxidative muscle fiber, few studies have investigated the effect of exercise mimetics on the release of myokines.

Eugenol is a safe natural compound extracted from clove oil and various plant spices such as basil, bay leaves, and cinnamon (*Kaur et al., 2010*). Eugenol exhibits a variety of biological activities, including antioxidative (*Magalhães et al., 2019*), antibacterial (*Devi et al., 2010*), anti-inflammatory (*Magalhães et al., 2019*), and anti-cancer activities (*Lesgards et al., 2014*). Recent studies have also suggested that eugenol could be a promising therapeutic drug for preventing diabetes and obesity (*Al Trad et al., 2019*; *Jung et al., 2012*; *Mnafgui et al., 2013*; *Srinivasan et al., 2014*), indicating its potential as an exercise mimetic. In addition, as a member of the transient receptor potential (TRP) channel family, TRP vanilloid 1 (TRPV1), also known as the capsaicin receptor, has been reported to improve endurance capacity and energy metabolism (*Luo et al., 2012*), counter obesity (*Baskaran et al., 2016*), and intervene diabetes (*Wang et al., 2012*), making it a potential target protein for discovering exercise mimetics.

Eugenol contains a vanilloyl fragment that may bind to TRPV1 similarly to capsaicin, implying that eugenol might exert its biological activity via TRPV1 (*Harb et al., 2019*). It has been found that eugenol activates TRPV1 in a heterologous expression system and rat trigeminal ganglion neurons (*Xu et al., 2006*; *Yang et al., 2003*). Furthermore, a recent study demonstrated that eugenol activated TRPV1 in skeletal muscle (*Jiang et al., 2022*). TRPV1 is an important $Ca^{2+}$ entry pathway contributing to increase intracellular $Ca^{2+}$ (*Nilius et al., 2007*), suggesting that TRPV1 may regulate many biological processes through $Ca^{2+}$-dependent signaling pathways. As a $Ca^{2+}$-dependent signaling, calcineurin (CaN) is a core signaling to promote fast-to-slow muscle fiber transformation (*Sakuma and Yamaguchi, 2010*). Additionally, CaN signaling plays an essential role in regulating the expression of myokine IL-6 (*Banzet et al., 2005*; *Banzet et al., 2007*). Therefore, we hypothesize that eugenol, as an exercise mimetic, promotes the release of myokines and fast-to-slow muscle fiber transformation via the TRPV1-mediated CaN signaling pathway. The primary objective of this study is to investigate this hypothesis.

## Results

### Eugenol promotes fast-to-slow muscle fiber transformation in mice and in C2C12 myotubes

As depicted in *Figure 1A and B*, eugenol did not have any effect on the body weight or skeletal muscle weight in mice. However, the skeletal muscle in the EUG50 and EUG100 groups exhibited a redder muscle color (*Figure 1C*), indicating a shift from fast-to-slow muscle fiber in these two groups. Further experiments showed that EUG50 and EUG100 increased the expression of slow MyHC and decreased the expression of fast MyHC protein in gastrocnemius (GAS) and tibialis anterior (TA) muscles (*Figure 1D and E*). Additionally, the mRNA expression of *Myh7*, *Myh2*, *Myh1*, and *Myh4* was generally consistent with protein expression (*Figure 1G*). As our in vivo studies demonstrated that the EUG200 group had no effect on muscle fiber type, we suspected that high doses of eugenol may have no impact on muscle fiber type. Therefore, we selected a broad range of eugenol doses (0–200 µM)

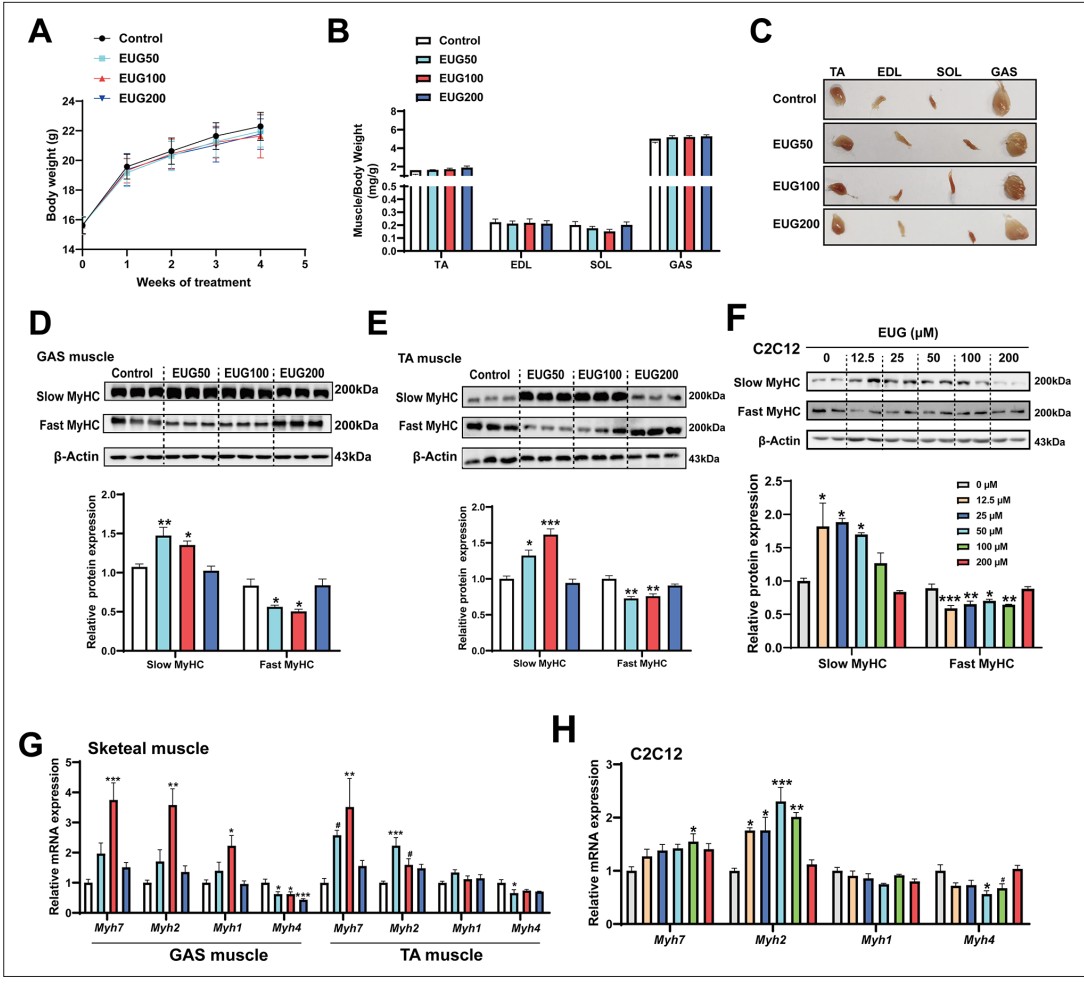

**Figure 1.** Eugenol promotes the transformation of fast-to-slow muscle fiber. (**A**) The body weight of the mice. (**B, C**) Skeletal muscle weight and representative images of skeletal muscle. (**D–F**) The protein expression of muscle fiber type in gastrocnemius (GAS) and tibialis anterior (TA) muscle and in C2C12 myotubes. (**G, H**) The mRNA expression of muscle fiber type in GAS and TA muscle and in C2C12 myotubes. For A, N=20 per group. For B, N=14 per group. For D and E, N=3 per group. For F and H, N=4 per group. For G, N=6 per group. One-way ANOVA test was used to determine statistical significance. *$p<0.05$, **$p<0.01$, and ***$p<0.001$.

The online version of this article includes the following source data and figure supplement(s) for figure 1:

**Source data 1.** The body weight of the mice (*Figure 1A*).

**Source data 2.** Skeletal muscle weight (*Figure 1B*).

**Source data 3.** Representative images of skeletal muscle (*Figure 1C*).

**Source data 4.** Original files for the western blot analysis (*Figure 1D*).

**Source data 5.** PDF containing *Figure 1D* and original scans of the relevant western blot analysis, with cropped areas.

**Source data 6.** Original files for the western blot analysis (*Figure 1E*).

**Source data 7.** PDF containing *Figure 1E* and original scans of the relevant western blot analysis, with cropped areas.

**Source data 8.** Original files for the western blot analysis (*Figure 1F*).

**Source data 9.** PDF containing *Figure 1F* and original scans of the relevant western blot analysis, with cropped areas.

**Source data 10.** The mRNA expression of muscle fiber type in gastrocnemius (GAS) and tibialis anterior (TA) muscle (*Figure 1G*).

**Source data 11.** The mRNA expression of muscle fiber type in C2C12 myotubes (*Figure 1H*).

*Figure 1 continued on next page*

*Figure 1 continued*

**Figure supplement 1.** Effect of eugenol on C2C12 cell viability.

**Figure supplement 1—source data 1.** The C2C12 cell viability was measured using CCK-8 kit.

based on the safe range of eugenol doses determined by the CCK-8 assay (*Figure 1—figure supplement 1*) to treat C2C12 myotubes and replicate these results in vivo. As indicated in *Figure 1F*, 12.5–50 μM eugenol boosted slow MyHC expression, while 12.5–100 μM eugenol decreased fast MyHC expression. Furthermore, as shown in *Figure 1H*, 100 μM eugenol increased the mRNA expression of *Myh7*, and 12.5–100 μM eugenol increased the mRNA expression of *Myh2*, whereas 50 μM eugenol decreased the mRNA expression of *Myh4*. Consistent with our in vivo studies, our in vitro findings again suggested that high doses (200 μM) of eugenol had no effect on muscle fiber type. In summary, our results suggest that eugenol promotes a transformation from fast-to-slow muscle fiber. However, it should be noted that high doses of eugenol may have no effect on muscle fiber type.

## Eugenol promotes oxidative metabolism activity, mitochondrial function, and endurance performance

An increase in the proportion of slow muscle fibers is often accompanied by an increase in skeletal muscle oxidative metabolism activity, mitochondrial function, and endurance performance. Therefore, we examined these indicators in our study. As shown in *Figure 2A*, EUG100 increased the exhaustion time of mice. As shown in *Figure 2B*, EUG50 and EUG100 decreased lactate dehydrogenase (LDH) activity and increased succinic dehydrogenase (SDH) activity in the GAS muscle. In the TA muscle, LDH activity was decreased in all EUG groups, and EUG100 increased the activities of SDH and malate dehydrogenase (MDH). Then, we chose 100 mg/kg eugenol (as the optimal dose) for the exhausting swimming test. In addition, the transcript levels of mitochondrial transcription factors *Pgc1a*, *Nrf1*, and *Tfam* and the mRNA expression of components of the mitochondrial electron transport chain were increased in eugenol-treated mice (*Figure 2C*). The protein expression of mitochondrial electron transport complex I and complex V was upregulated in eugenol-treated mice (*Figure 2D*). Furthermore, EUG100 improved the mtDNA copy number in the GAS and TA muscle (*Figure 2E*), and 12.5–100 μM EUG increased the mtDNA copy number in C2C12 myotubes (*Figure 2F*).

## Eugenol promotes lipolysis and browning of fat

Our research has found that both EUG100 and EUG200 promoted average daily feed intake (ADF) (*Figure 3A*). However, interestingly, there was no change in body weight (*Figure 1A*). This indicates that there was an increase in ADF/average daily weight gain (ADG) (*Figure 3A*). In addition, eugenol decreased inguinal white adipose tissue (iWAT) and gonadal white adipose tissue (gWAT) weight while promoting brown adipose tissue (BAT) weight (*Figure 3B*). And eugenol decreased T-CHO and LDL while increasing HDL level in serum (*Figure 3C*). These apparent results suggest that eugenol may increase the energy metabolism rate in mice, offsetting the weight gain and increased fat synthesis resulting from increased food intake. Therefore, we further speculate that eugenol may promote lipolysis and fat thermogenesis. As we speculated, we found that eugenol promoted the mRNA expression of the fat synthesis-related genes *Pparg* and *Hsl*, as well as the fatty acid transport gene *Fabp4*, in iWAT (*Figure 3D*). In gWAT, it was found that eugenol promoted the mRNA expression of the *Fasn* and *Fabp4* (*Figure 3D*). We further examined the mRNA expression of key genes involved in browning of fat, and found that eugenol promoted the mRNA expression of *Cd137*, *Tbx1*, *Ucp1*, *Prdm16*, *Dio2*, and *Cidea* in iWAT, while eugenol promoted the mRNA expression of *Tmem26*, *Ucp1*, *Prdm16*, *Dio2*, and *Cidea* in gWAT (*Figure 3E*). At the protein level, eugenol promoted the expression of the fatty acid transport protein FABP-1, as well as the UCP-1 protein expression in gWAT (*Figure 3F*). In addition, we also examined the effects of eugenol on the browning-related proteins and mitochondrial complex proteins in BAT. It was found that eugenol promoted the UCP-1 and PGC-1α protein expression (*Figure 3G*), as well as the mitochondrial electron transport complex III and complex V protein expression (*Figure 3H*). Together, these results indicate that eugenol promotes lipolysis and browning of fat.

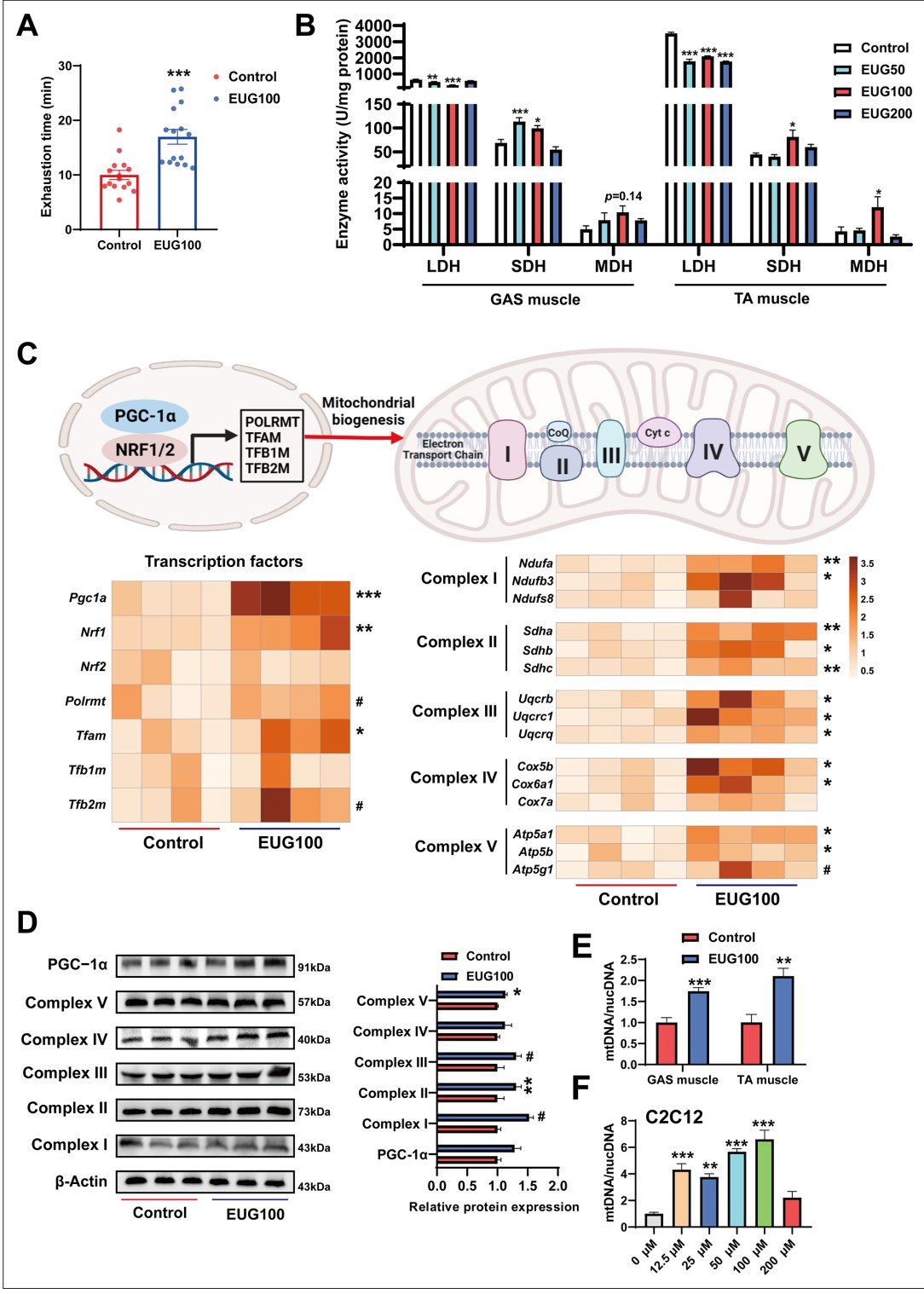

**Figure 2.** Eugenol promotes oxidative metabolism activity, mitochondrial function, and endurance performance of mice. (**A**) The effect of eugenol on exhausting swimming time in mice. (**B**) The effect of eugenol on metabolism enzymes activity in gastrocnemius (GAS) and tibialis anterior (TA) muscle. (**C**) The heatmap for the mRNA expression of genes encoding mitochondrial complex components and transcription factors controlling mitochondrial biogenesis in GAS muscle. Color gradient represents relative mRNA expression with darker colors indicating higher expression. (**D**) Protein expression of mitochondrial electron transport complexes in GAS muscle. Complex I (NDUFA1), complex II (SDHA), complex III (UQCRC1), complex IV (MTCO1), and complex V (ATP5B). (**E**)

*Figure 2 continued on next page*

*Figure 2 continued*

mtDNA copy number in muscles. (**F**) mtDNA copy number in C2C12 myotubes. For A, N=15 per group. For B, N=6 per group. For C and F, N=4 per group. For D, N=3 per group. For E, N=6 per group. One-way ANOVA test was used to determine statistical significance for B and F, student's *t*-test was used to determine statistical significance for other panels. #$p<0.1$, *$p<0.05$, **$p<0.01$, and ***$p<0.001$.

The online version of this article includes the following source data for figure 2:

**Source data 1.** Exhausting swimming time in mice (*Figure 2A*).

**Source data 2.** Metabolism enzymes activity in gastrocnemius (GAS) and tibialis anterior (TA) muscle (*Figure 2B*).

**Source data 3.** The mRNA expression of genes encoding mitochondrial complex components and transcription factors controlling mitochondrial biogenesis in gastrocnemius (GAS) muscle (*Figure 2C*).

**Source data 4.** Original files for the western blot analysis (*Figure 2D*).

**Source data 5.** PDF containing *Figure 2D* and original scans of the relevant western blot analysis, with cropped areas.

**Source data 6.** mtDNA copy number in muscles (*Figure 2E*).

**Source data 7.** mtDNA copy number in C2C12 myotubes (*Figure 2F*).

## Eugenol activates TRPV1-mediated CaN/NFATc1 signaling pathway in skeletal muscle

Based on the gene expression profiling, it was observed that TRPV1 is expressed in all tissues, including adipose and muscle tissues, with the highest expression in skeletal muscle compared to other TRP channels (*Figure 4A* and *Figure 4—figure supplement 1A*). Quantitative PCR (qPCR) analysis confirmed the expression of TRPV1 and TRPV2 in skeletal muscle, and the mRNA expression of only *Trpv1* was promoted by EUG50 and EUG100 in TA muscle. In C2C12 cells, only TRPV1-4 were expressed, and only *Trpv1* mRNA expression was promoted by 25 and 50 µM eugenol (*Figure 4C*). Adipose tissue expressed TRPV1 and TRPV2 genes, and EUG100 and EUG200 promoted *Trpv1* mRNA expression, while EUG50 promoted *Trpv2* mRNA expression (*Figure 4—figure supplement 1B and C*). The effect of eugenol on TRPV1 protein expression was consistent with its effect on *Trpv1* mRNA expression in both skeletal muscle tissue and C2C12 cells (*Figure 4D–F*). Moreover, taking the TRPV1-capsaicin binding sites (TYR511, SER512, THR550, and GLU570) (*Carnevale and Rohacs, 2016*) as the potential binding pocket, molecular docking analysis showed that eugenol bound at the binding pocket and interacted with THR550, ASN551, LEU553, TYR554, ALA566, ILE569, GLU570, and ILE573 (*Figure 4—figure supplement 2*). Based on the above results, we conclude that eugenol has the potential to activate TRPV1 in both skeletal muscle and adipose tissue. We further investigated the TRPV1-mediated CaN/NFATc1 signaling pathway in skeletal muscle. CnA is a catalytic subunit of CaN, and its expression level reflects the activity of CaN. Our results showed that EUG50 and EUG100 increased CnA protein expression in the GAS and TA muscles (*Figure 4D and E*), and 12.5–100 µM eugenol promoted CnA protein expression in C2C12 myotubes (*Figure 4F*). The regulator of calcineurin 1 (MCIP1) is a biomarker to reflect the CaN activity (*Yang et al., 2000*), the mRNA expression of *Mcip1* was increased in EUG100 groups (*Figure 4—figure supplement 3*). Furthermore, the nuclear translocation of NFATc1 was promoted by EUG50 and EUG100 in the GAS and TA muscles, and by 12.5–50 µM eugenol in C2C12 myotubes (*Figure 4G–I*). In summary, our in vivo and in vitro studies both suggest that eugenol can activate the TRPV1-mediated CaN/NFATc1 signaling pathway in skeletal muscle. Interestingly, similar to the effects of eugenol on muscle fiber types, we again found that high doses of eugenol have no effect on the signaling pathway.

## Eugenol promotes fast-to-slow muscle fiber transformation by activating TRPV1-mediated CaN/NFATc1 signaling pathway

To further investigate the role of TRPV1 in regulating fast-to-slow muscle fibers, we treated C2C12 myotubes with 25 µM eugenol and either 1 µM TRPV1 inhibitor AMG-517 or 0.5 µM CaN inhibitor cyclosporine A (CsA). The results showed that TRPV1 inhibition weakened the increase in intracellular $Ca^{2+}$ levels induced by eugenol, suggesting that eugenol acts via TRPV1 (*Figure 5A*). Furthermore, the inhibition of TRPV1 and CaN attenuated the effect of eugenol on CaN (*Figure 5B*). In addition, eugenol increased the mitochondrial electron transport complex I, II, III, and V protein expression, the

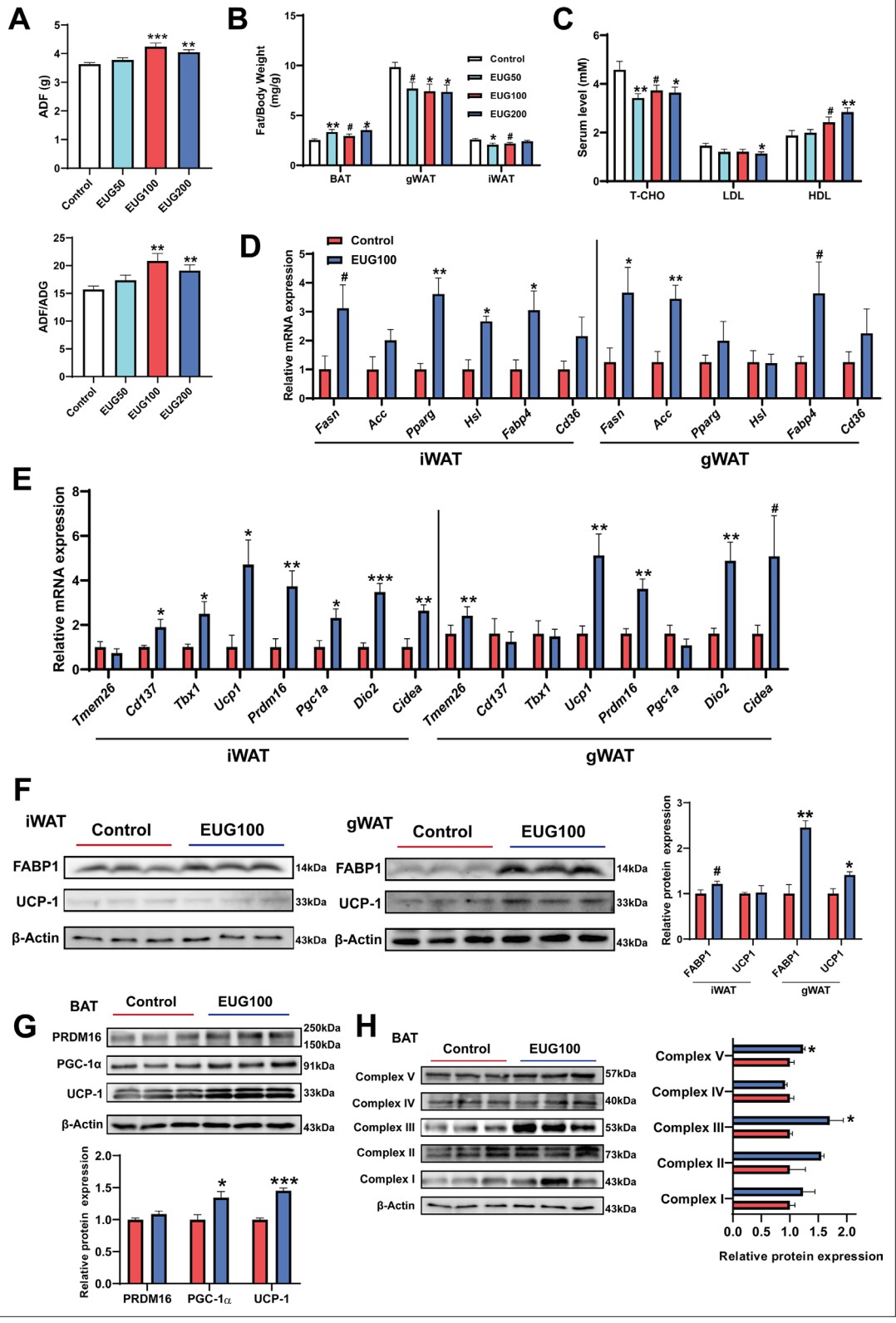

**Figure 3.** Eugenol enhances lipolysis and thermogenesis. (**A**) The average daily feed intake (ADF) and the ration of ADF to average daily weight gain (ADG). (**B**) Tissue weight of adipose weight. (**C**) The level of T-CHO, LDL, and HDL in serum. (**D**) The mRNA expression of genes related to lipolysis, lipogenesis, and lipid transport in inguinal white adipose tissue (iWAT) and gonadal white adipose tissue (gWAT). (**E**) The mRNA expression of genes related

*Figure 3 continued on next page*

*Figure 3 continued*

to adipose browning and thermogenesis in iWAT and gWAT. (**F**) The protein expression of FABP1 and UCP1 in iWAT and gWAT. (**G**) The expression of protein related to adipose browning and thermogenesis in brown adipose tissue (BAT). (**H**) The protein expression of mitochondrial electron transport complexes in BAT. For A, N=20 per group. For B, N=14 per group. For C, N=8 per group. For D and E, N=6 per group. For F and H, N=3 per group. One-way ANOVA test was used to determine statistical significance for A-C, student's *t*-test was used to determine statistical significance for other panels. #$p<0.1$, *$p<0.05$, **$p<0.01$, and ***$p<0.001$.

The online version of this article includes the following source data for figure 3:

**Source data 1.** The average daily feed intake (ADF) and the ration of ADF to average daily weight gain (ADG) (*Figure 3A*).

**Source data 2.** Tissue weight of adipose weight (*Figure 3B*).

**Source data 3.** The level of T-CHO, LDL, and HDL in serum (*Figure 3C*).

**Source data 4.** The mRNA expression of genes related to lipolysis, lipogenesis, and lipid transport in inguinal white adipose tissue (iWAT) and gonadal white adipose tissue (gWAT) (*Figure 3D*).

**Source data 5.** The mRNA expression of genes related to adipose browning and thermogenesis in inguinal white adipose tissue (iWAT) and gonadal white adipose tissue (gWAT) (*Figure 3E*).

**Source data 6.** Original files for the western blot analysis (*Figure 3F*).

**Source data 7.** PDF containing *Figure 3F* and original scans of the relevant western blot analysis, with cropped areas.

**Source data 8.** Original files for the western blot analysis (*Figure 3G*).

**Source data 9.** PDF containing *Figure 3G* and original scans of the relevant western blot analysis, with cropped areas.

**Source data 10.** Original files for the western blot analysis (*Figure 3H*).

**Source data 11.** PDF containing *Figure 3H* and original scans of the relevant western blot analysis, with cropped areas.

---

inhibition of TRPV1 and CaN attenuated the effect of eugenol on complex I, III, and V (*Figure 5C*). Importantly, immunofluorescence and western blot analysis revealed that eugenol promoted the fast-to-slow muscle fiber transformation, while the inhibition of TRPV1 and CaN eliminated this effect (*Figure 5D* and *Figure 5E*).

## The myokines regulated by CaN

To investigate which myokines are controlled by CaN, C2C12 myotubes were treated with the $Ca^{2+}$ ionophore A23187. As shown in *Figure 6A–C*, A23187 increased the mRNA expression of *MCIP1*, the protein expression of CnA, and CaN activity. We chose 0.5 µM A23187 for subsequent experiments and confirmed that it increased intracellular $Ca^{2+}$ levels (*Figure 6D*). To identify CaN-controlled myokines, we next detected the mRNA expression of several myokines that have been well documented to improve metabolic homeostasis (*Eckel, 2019*; *Whitham and Febbraio, 2016*) and promote fast-to-slow muscle fiber transformation (*Correia et al., 2021*; *Knudsen et al., 2020*; *Men et al., 2021*; *Quinn et al., 2013*; *Table 1*). As shown in *Figure 6E*, A23187 significantly increased the mRNA expression of *Fndc5*, *Il6*, *Il15*, and *Metrnl*. However, *Il13* mRNA expression was not detected in C2C12 myotubes.

**Table 1.** Myokines to be tested in our study.

| | Myokines | References |
|---|---|---|
| Myokines that improve metabolic homeostasis | CX3CL1, FGF21, FNDC5, IL-6, IL-8, IL-15 | *Whitham and Febbraio, 2016* |
| | Metrnl, FGF21, FNDC5, Myonectin (Erfe) | *Eckel, 2019* |
| Myokines that improve endurance capacity and promote fast-to-slow muscle fiber transformation | FNDC5 | *Men et al., 2021* |
| | IL-13 | *Knudsen et al., 2020* |
| | IL-15 | *Quinn et al., 2013* |
| | Neurturin (Nrtn) | *Correia et al., 2021* |

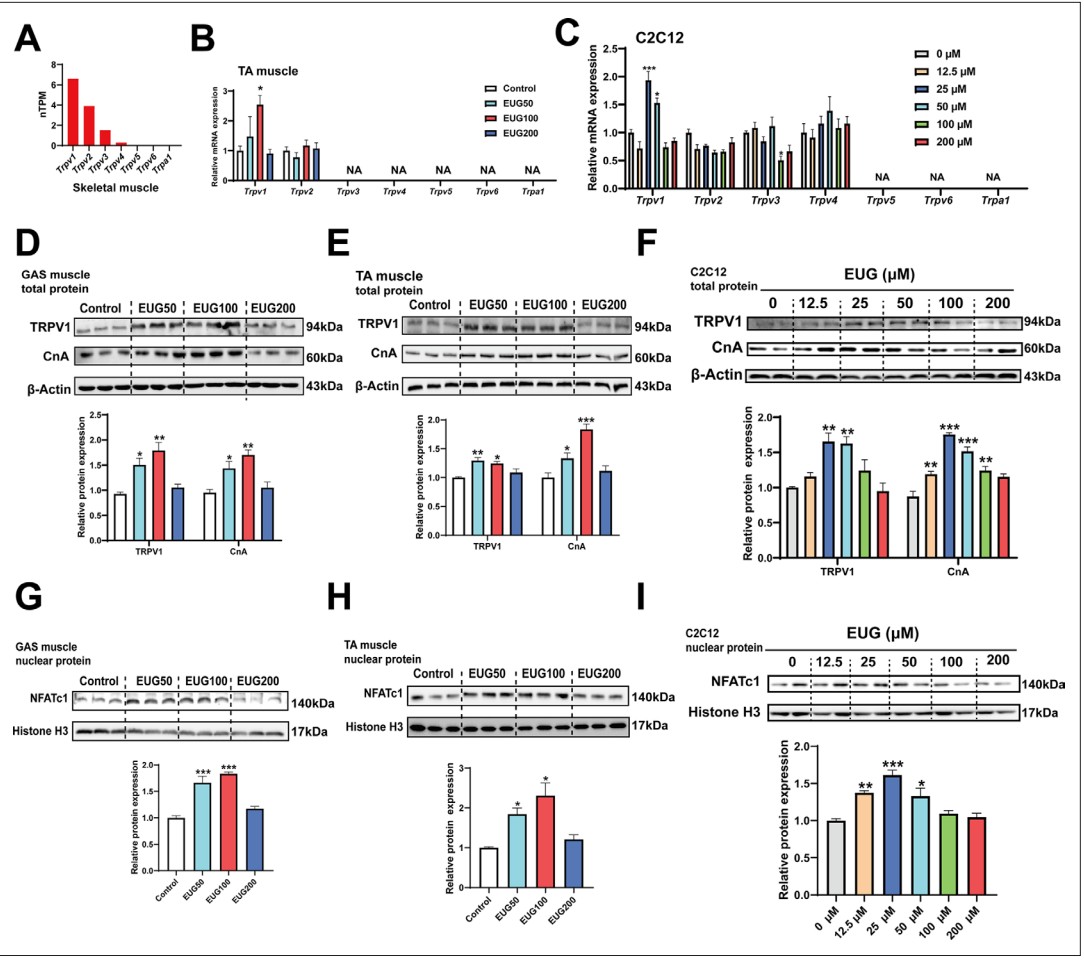

**Figure 4.** Eugenol activated TRPV1-mediated CaN/NFATc1 signaling pathway. (**A**) The gene expression profile of transient receptor potential (TRP) channels in skeletal muscle was obtained from the GTEx dataset in The Human Protein Atlas (https://www.proteinatlas.org/). (**B, C**) The mRNA expression of TRP channels in tibialis anterior (TA) muscle and C2C12 myotubes. (**D–F**) The TRPV1 and CnA protein expression in gastrocnemius (GAS) and TA muscle and in C2C12 myotubes. (**G–I**) The protein expression of NFATc1 in GAS and TA muscle and in C2C12 myotubes. For B, N=6 per group. For C, F, and I, N=4 per group. For D, E and G, H, N=3 per group. One-way ANOVA test was used to determine statistical significance. *p<0.05, **p<0.01, and ***p<0.001.

The online version of this article includes the following source data and figure supplement(s) for figure 4:

**Source data 1.** The gene expression profile of transient receptor potential (TRP) channels in skeletal muscle (*Figure 4A*).

**Source data 2.** The mRNA expression of transient receptor potential (TRP) channels in tibialis anterior (TA) muscle (*Figure 4B*).

**Source data 3.** The mRNA expression of transient receptor potential (TRP) channels in C2C12 myotubes (*Figure 4C*).

**Source data 4.** Original files for the western blot analysis (*Figure 4D*).

**Source data 5.** PDF containing *Figure 4D* and original scans of the relevant western blot analysis, with cropped areas.

**Source data 6.** Original files for the western blot analysis (*Figure 4E*).

**Source data 7.** PDF containing *Figure 4E* and original scans of the relevant western blot analysis, with cropped areas.

**Source data 8.** Original files for the western blot analysis (*Figure 4F*).

**Source data 9.** PDF containing *Figure 4F* and original scans of the relevant western blot analysis, with cropped areas.

*Figure 4 continued on next page*

*Figure 4 continued*

**Source data 10.** Original files for the western blot analysis (*Figure 4G*).

**Source data 11.** PDF containing *Figure 4G* and original scans of the relevant western blot analysis, with cropped areas.

**Source data 12.** Original files for the western blot analysis (*Figure 4H*).

**Source data 13.** PDF containing *Figure 4H* and original scans of the relevant western blot analysis, with cropped areas.

**Source data 14.** Original files for the western blot analysis (*Figure 4I*).

**Source data 15.** PDF containing *Figure 4I* and original scans of the relevant western blot analysis, with cropped areas.

**Figure supplement 1.** Transient receptor potential (TRP) channels expression profiles and *Trpv1* mRNA expression in adipose tissue.

**Figure supplement 1—source data 1.** Transient receptor potential (TRP) channels expression profiles in adipose tissue (*Figure 4—figure supplement 1B*).

**Figure supplement 1—source data 2.** The mRNA expression of transient receptor potential (TRP) channels in adipose tissue (*Figure 4—figure supplement 1C*).

**Figure supplement 2.** Molecular docking for eugenol and TRPV1.

**Figure supplement 3.** The mRNA expression of *Mcip1*.

**Figure supplement 3—source data 1.** The mRNA expression of *Mcip1*.

---

In addition, a correlation heatmap showed the highest correlation coefficient (R=0.954) between *Mcip1* expression and *Il15* expression, followed by *Mcip1* and *Il6* (R=0.846), *Mcip1* and *Metrnl* (0.77), and *Mcip1* and *Fndc5* (0.651) (*Figure 6F*). Furthermore, a TFBS prediction revealed more potential binding sites between the *Il15* promoter and NFATc1 (*Figure 6G*). Based on the correlation and TFBS analysis, it was suggested that IL-15 is the myokine most likely to be regulated by the CaN/NFATc1 signaling pathway, and we selected it for further study.

## The myokine IL-15 expression depends on CaN/NFATc1 signaling pathway

Firstly, it was observed that treatment with 0.5 and 1 µM $Ca^{2+}$ ionophore led to upregulation of IL-15 protein expression (*Figure 7A*). Subsequently, we treated C2C12 myotubes with 0.5 µM $Ca^{2+}$ ionophore and 0.5 µM CsA to investigate their effects. We found that $Ca^{2+}$ ionophore treatment upregulated the expression of CnA, NFATc1, and IL-15 proteins, while inhibition of CaN eliminated these effects (*Figure 7B–D*). Furthermore, $Ca^{2+}$ ionophore treatment increased the expression of slow MyHC and decreased the expression of fast MyHC, while CsA blocked this effect (*Figure 7— figure supplement 1*). Based on the transcription factor motif databases (JASPAR and hIFtarget), it was predicted that a sequence (5'-AATGGAAAA-3') in the promoter regions of IL-15 was a potential binding site of NFATc1 (*Figure 7E*), and DNA-protein docking analysis also revealed a high probability of binding between this sequence and the NFATc1 protein (*Figure 7F*). We then performed an electrophoretic mobility shift assay (EMSA) to validate the binding of NFATc1 to this sequence. The probes used in the EMSA were shown in *Figure 7G* and the EMSA results were shown in *Figure 7H*. We observed upward migration bands when the C2C12 nuclear protein extract (NE) was incubated with the bio-NFATc1 probe (lanes 2–4), and compared to the control (lane 2) and CsA inhibition group (lane 4), $Ca^{2+}$ treatment group showed higher expression of the band (lane 3). Moreover, the use of competing cold-NFATc1 probes showed no upward migration band (lane 5), while using the mut-NFATc1 probe again showed the band (lane 6). These findings suggest that NFATc1 may bind to the promoter of IL-15. We further investigated whether NFATc1 transcriptionally activates IL-15 through luciferase reporter assays. As shown in *Figure 7I*, overexpression of NFATc1 promoted NFATc1 protein expression in HEK293T cells. Additionally, after transfection with the IL-15 reporter plasmid, overexpression of NFATc1 enhanced the relative fluorescence intensity (*Figure 7J*), suggesting that the transcriptional activation of IL-15 is regulated by NFATc1.

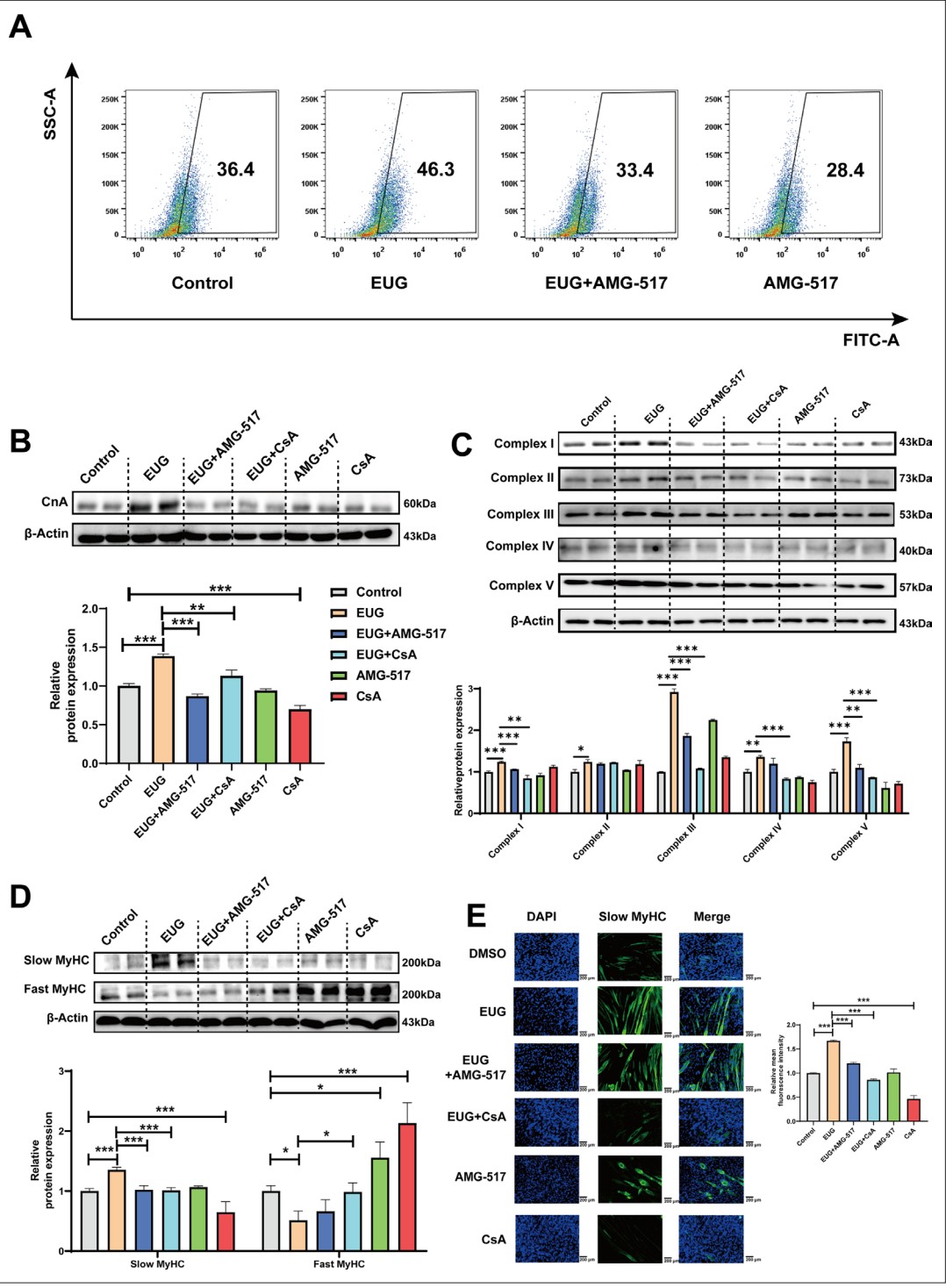

**Figure 5.** Eugenol promotes fast-to-slow muscle fiber transformation by activating TRPV1-mediated CaN/NFATc1 signaling pathway. C2C12 myotubes were treated by 25 µM eugenol and 1 µM TRPV1 inhibitor AMG-517 or 0.5 µM CaN inhibitor cyclosporine A (CsA) for 1 day after 4 days of differentiation. (**A**) The flow cytometry assay was used to detect $Ca^{2+}$ levels in C2C12 myotubes; FITC means the fluo-4 fluorescence and SSC means side scatter. (**B**) Western blot was used to detect CnA protein expression in C2C12 myotubes. (**C**) Western blot was used to detect mitochondrial electron transport complexes protein expression in C2C12 myotubes. (**D**) Western blot was used to detect slow myosin heavy chain (MyHC) and fast MyHC protein expression in C2C12 myotubes. (**E**) Representative immunofluorescence images of slow MyHC (green fluorescence) and relative mean fluorescence intensity

*Figure 5 continued on next page*

*Figure 5 continued*

quantification. Magnification: ×200. For A–D, N=4 per group. One-way ANOVA test was used to determine statistical significance. *p<0.05, **p<0.01, and ***p<0.001.

The online version of this article includes the following source data for figure 5:

**Source data 1.** The flow cytometry assay (*Figure 5A*).

**Source data 2.** Original files for the western blot analysis (*Figure 5B*).

**Source data 3.** PDF containing *Figure 5B* and original scans of the relevant western blot analysis, with cropped areas.

**Source data 4.** Original files for the western blot analysis (*Figure 5C*).

**Source data 5.** PDF containing *Figure 5C* and original scans of the relevant western blot analysis, with cropped areas.

**Source data 6.** Original files for the western blot analysis (*Figure 5D*).

**Source data 7.** PDF containing *Figure 5D* and original scans of the relevant western blot analysis, with cropped areas.

**Source data 8.** Representative immunofluorescence images of slow myosin heavy chain (MyHC) (*Figure 5E*).

## Eugenol promotes IL-15 level by TRPV1-mediated CaN/NFATc1 signaling pathway

We conducted further experiments to examine the effect of eugenol on IL-15 expression. Our results (*Figure 8A–C*) showed that EUG50 and EUG100 promoted the mRNA and protein expression of IL-15 in both GAS and TA muscle. Interestingly, in extensor digitorum longus (EDL) and soleus (SOL) muscle, which are dominated by fast and slow muscle fibers respectively, EUG100 promoted *Il15* mRNA expression (*Figure 8D*). We also found that the mRNA expression of *Il15* was higher in SOL muscle than in EDL muscle (*Figure 8D*). Moreover, Pearson's correlation analysis showed that *Il15* expression positively correlated with *Myh7* (R=0.714) and *Myh2* (R=0.774) expression, and negatively correlated with *Myh4* (R=−0.568) (*Figure 8D*). Consistent with the mRNA expression data, the IL-15 protein expression was also higher in SOL muscle than in EDL muscle (*Figure 8E*). Additionally, EUG50 and EUG100 increased the concentration of IL-15 in the serum (*Figure 8F*). These findings suggested that IL-15 was a oxidative muscle fiber type-specific myokine that was promoted by eugenol. Our in vitro experiments showed that 25 and 50 µM eugenol increased the mRNA expression and secretion of IL-15 (*Figure 9A and B*), which was consistent with our in vivo experiments. In addition, the inhibition of TRPV1 and CaN decreased the upregulation of eugenol on IL-15 mRNA and protein expression (*Figure 9C and D*). Immunofluorescence staining with IL-15 also showed similar results (*Figure 9—figure supplement 1*).

## Discussion

Our study reveals that eugenol can serve as an exercise mimetic that promotes remodeling of skeletal muscle fibers and expression of the myokine IL-15 through the TRPV1-mediated CaN/NFATc1 signaling pathway. This expands the traditional biological functions of eugenol and provides a theoretical basis for its potential applications in the food and drug industry. Moreover, we identify a novel TRPV1-mediated CaN/NFATc1 signaling pathway that promotes the transformation of fast-to-slow muscle fibers. Importantly, we provide the first explanation of the mechanism underlying the release of the myokine IL-15.

TRPV1, like all other TRP channels, assembles as tetramers to form cation-permeable pores (*Clapham, 2003*). Initially identified as a capsaicin receptor and heat-activated ion channel that modulates pain and neurogenic inflammation (*Caterina et al., 1997*), subsequent studies have found that TRPV1 is expressed on many non-neural sites and plays roles in immunity, vasculature, obesity, and thermogenesis (*Fernandes et al., 2012*). Previous studies have also shown that capsaicin activated TRPV1 to improve endurance capacity and energy metabolism (*Luo et al., 2012*), counter obesity (*Baskaran et al., 2016*), and intervene diabetes (*Wang et al., 2012*). Therefore, TRPV1 is a strong target for the discovery of exercise mimetics, which drove us to search for TRPV1 agonists that may act as exercise mimetics. We focused on eugenol, a healthy and edible plant extract that may be a

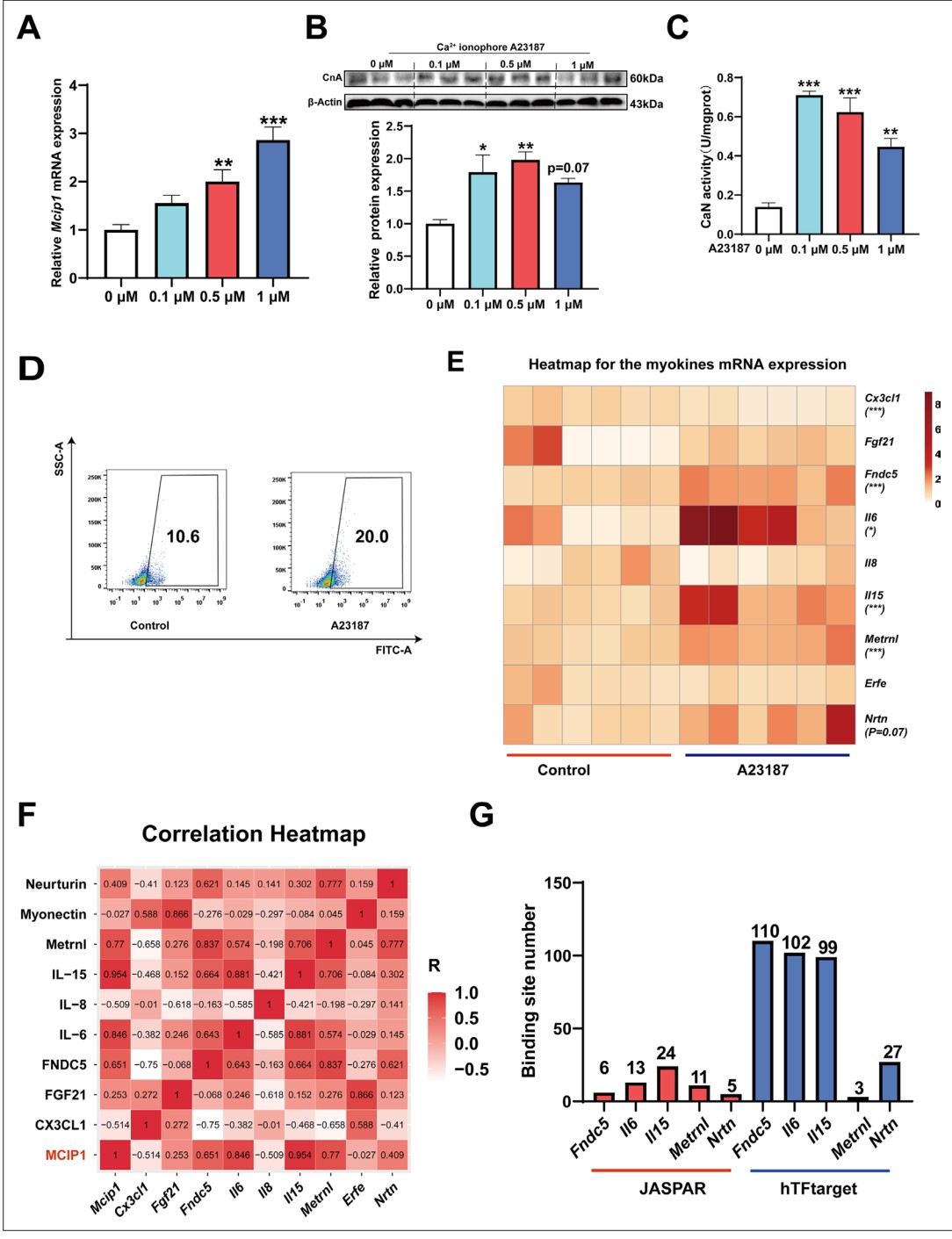

**Figure 6.** The myokines controlled by calcineurin (CaN). C2C12 myotubes were treated for 16 hr with 0, 0.1, 0.5, and 1 μM Ca$^{2+}$ ionophore after 2 days of differentiation. (**A**) The mRNA expression of *Mcip1*. (**B**) The protein expression of CnA. (**C**) The enzyme activity of CaN. (**D**) Fluo-4 was used to stain the Ca$^{2+}$ and the flow cytometry assay was used to detect Ca$^{2+}$ fluorescence in C2C12 myotubes in control and 0.5 μM A23187 groups. (**E**) The heatmap for the myokines mRNA expression in control and 0.5 μM A23187 groups. Color gradient represents relative mRNA expression with darker colors indicating higher expression. (**F**) Correlation analysis of gene expression values of myokines and MCIP1 gene performed by linear regression with Pearson's correlation coefficient (r). Color gradient represents correlation coefficient with darker colors indicating higher positive correlation. (**G**) The number of binding sites for transcription factors NFATc1 were predicted by hTFtarget and JASPAR. For A, N=6 per group. For B and C, N=3 per group. For D, N=4 per group. For E, N=6 per group. One-

*Figure 6 continued on next page*

*Figure 6 continued*

way ANOVA test was used to determine statistical significance for A-C, student's *t*-test was used to determine statistical significance for other panels. *p<0.05, **p<0.01, and ***p<0.001.

The online version of this article includes the following source data for figure 6:

**Source data 1.** The mRNA expression of *Mcip1* (*Figure 6A*).

**Source data 2.** Original files for the western blot analysis (*Figure 6B*).

**Source data 3.** PDF containing *Figure 6B* and original scans of the relevant western blot analysis, with cropped areas.

**Source data 4.** The enzyme activity of calcineurin (CaN) (*Figure 6C*).

**Source data 5.** The flow cytometry assay (*Figure 6D*).

**Source data 6.** The myokines mRNA expression (*Figure 6E*).

**Source data 7.** Correlation analysis of gene expression values (*Figure 6F*).

potential TRPV1 agonist. Since eugenol contains a vanilloyl fragment, it is possible to bind TRPV1 through a similar pattern to capsaicin. Indeed, previous studies have found that eugenol activates TRPV1 in a heterologous expression system and rat trigeminal ganglion neurons (*Xu et al., 2006*; *Yang et al., 2003*), and a recent study reported that 50 and 100 µM eugenol promoted TRPV1 expression in C2C12 myotubes (*Jiang et al., 2022*). In our study, we investigated the mRNA expression profile of TRP channels in C2C12 myotubes and skeletal muscle, and found that only TRPV1 mRNA was upregulated in response to eugenol treatment. Moreover, it was observed that the protein expression of TRPV1 was consistent with the mRNA expression. Our molecular docking results also indicated that eugenol bound to the capsaicin binding pockets of TRPV1. Based on these findings, we propose that eugenol specifically activates TRPV1 in skeletal muscle, rather than other TRP channels. This may be due to the relatively higher expression of TRPV1 in skeletal muscle compared to other TRP channels.

One of the benefits of exercise mimetics for body is to promote the skeletal muscle oxidative phenotype and endurance performance (*Fan and Evans, 2017*). Our studies showed that eugenol improved endurance performance and promoted fast-to-slow muscle fiber transformation. Slow muscle fibers are characterized by the improvement in mitochondrial function and oxidative metabolism (*Choi and Kim, 2009*). As expected, eugenol also promoted mitochondrial function and oxidative metabolism capacity in skeletal muscle. In a previous study, capsaicin-induced TRPV1 activation was shown to promote exercise endurance and the skeletal muscle oxidative phenotype (*Wang et al., 2012*). The authors suggested that TRPV1 activated calmodulin-dependent protein kinase (CaMK) by increasing intracellular Ca$^{2+}$, and the activation of CaMK further activated PGC-1α, contributing to these effects (*Wang et al., 2012*). In skeletal muscle, the CaN signaling pathway is also an important Ca$^{2+}$-mediated signal that promotes the muscle oxidative phenotype (*Sakuma and Yamaguchi, 2010*). CaN induces the translocation of NFATc1 to the nucleus by dephosphorylating NFATc1, thereby switching fast-to-slow muscle fibers (*Calabria et al., 2009*). TRPV1 activation has been shown to promote CaN activity in several other cells and tissues (*Hou et al., 2019*; *Ma et al., 2011*; *Yang et al., 2018*), but the link between TRPV1 and CaN signaling in skeletal muscle has not yet been established, nor has it been determined whether TRPV1 promotes muscle oxidative phenotype through the CaN/NFATc1 signaling pathway. Therefore, we investigated the role of TRPV1-mediated CaN signaling pathway in the regulation of muscle fiber types. We found that eugenol increased CaN expression through the activation of TRPV1, and that it promoted fast-to-slow muscle fiber transformation via the TRPV1-mediated CaN/NFATc1 signaling pathway. Recent studies have also found that eugenol increases muscle glucose uptake through the TRPV1/CaMK signaling pathway (*Jiang et al., 2022*), suggesting that both CaN and CaMK signaling may be involved in TRPV1-facilitated skeletal muscle oxidative phenotype.

Another benefit of exercise mimetics is the alleviation of obesity and improvement of metabolic health. It has been reported that an increase in slow-twitch fiber content is positively associated with reduced obesity and improved metabolic health (*Carlson et al., 2010*). Previous studies have shown that eugenol can reduce blood lipids in rats with hyperlipidemia (*Harb et al., 2019*), lower blood glucose and insulin resistance in diabetic mice (*Al Trad et al., 2019*; *Sanae et al., 2014*), and reduce hepatic lipid accumulation in high-fat-fed mice (*Rodrigues et al., 2022*), demonstrating its potential in improving glucose and lipid metabolism. In our study, we found that feeding eugenol under standard

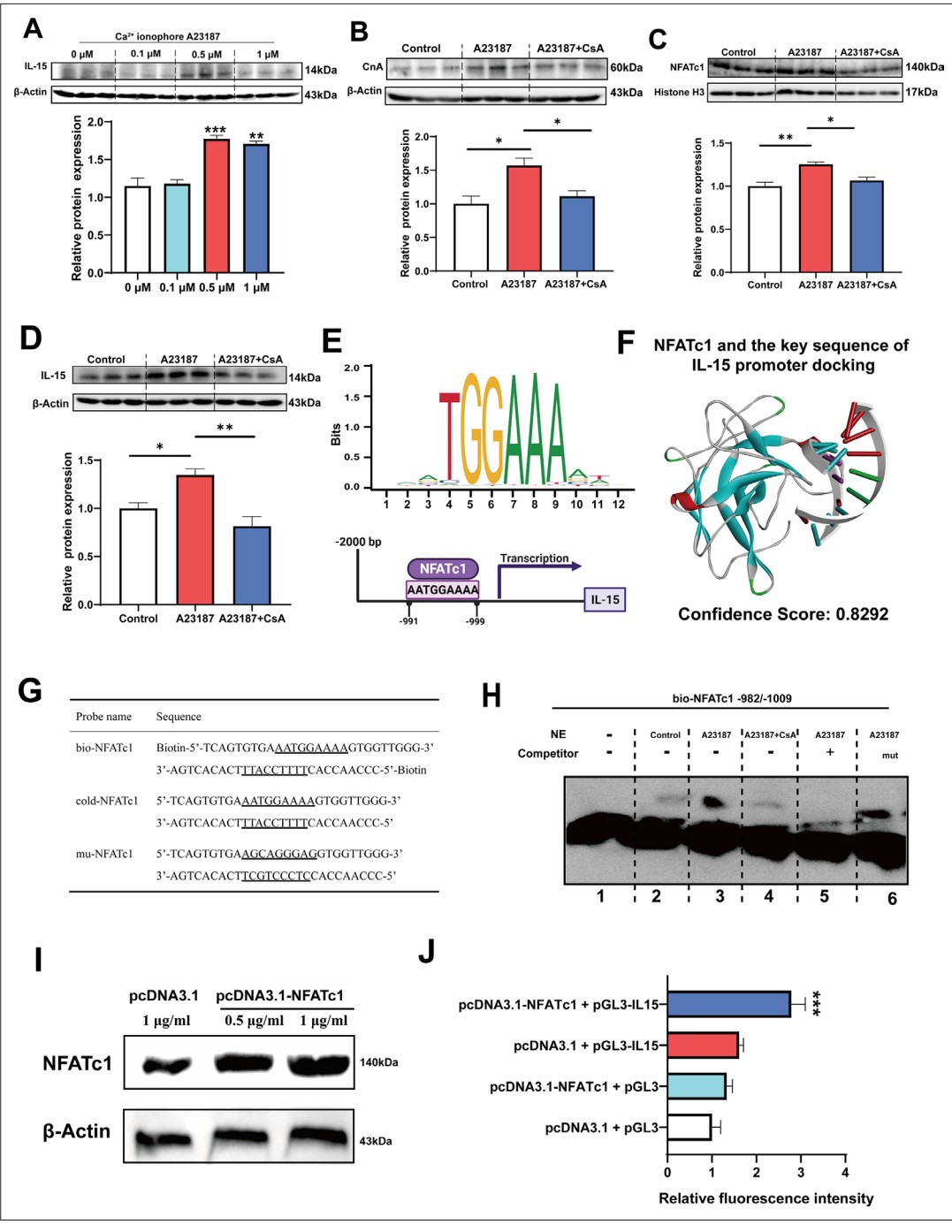

**Figure 7.** The myokine IL-15 expression depends on CaN/NFATc1 signaling pathway. (**A**) C2C12 myotubes were treated for 16 hr with 0, 0.1, 0.5, and 1 μM Ca²⁺ ionophore after 2 days of differentiation. The protein expression of IL-15. (**B–D**) C2C12 myotubes were treated by 0.5 μM A23187 and 0.5 μM cyclosporine A (CsA) for 16 hr after 2 days of differentiation. The protein expression of CnA, NFATc1, and IL-15. (**E**) Sequence logo of NFATc1 motif and the predicted NFATc1 binding sites in the promoter region of IL-15. (**F**) NFATc1 and the key sequence of IL-15 (5'-AATGAAAA-3') docking. Confidence scores above 0.7 indicate high probability of binding, scores between 0.5 and 0.7 suggest possible binding, and scores below 0.5 indicate unlikely binding. (**G**) The probe sequence of NFATc1. The underline represents the predicted binding site of NFATc1, bio-NFATc1 means oligonucleotide probes that labeled with biotin at the 5' end, cold-NFATc1 means oligonucleotide probes that did not label with biotin, mu-NFATc1 means oligonucleotide probes that were mutated at the binding site. (**H**) Nuclear protein extracts (NE) with NFATc1 probe were used for electrophoretic mobility shift assay (EMSA). (**I**) The protein expression of NFATc1 after transfecting 1 μg/mL pcDNA3.1 vector, 0.5 μg/mL or 1 μg/mL pcDNA3.1-NFATc1 in

*Figure 7 continued on next page*

*Figure 7 continued*

HEK293T cells. (**J**) The relative luciferase intensity referred to the ratio between firefly luciferase intensity and Renilla luciferase intensity. For A–D, N=3 per group. For J, N=6 per group. One-way ANOVA test was used to determine statistical significance. $*p<0.05$, $**p<0.01$, and $***p<0.001$.

The online version of this article includes the following source data and figure supplement(s) for figure 7:

**Source data 1.** Original files for the western blot analysis (*Figure 7A*).

**Source data 2.** PDF containing *Figure 7A* and original scans of the relevant western blot analysis, with cropped areas.

**Source data 3.** Original files for the western blot analysis (*Figure 7B*).

**Source data 4.** PDF containing *Figure 7B* and original scans of the relevant western blot analysis, with cropped areas.

**Source data 5.** Original files for the western blot analysis (*Figure 7C*).

**Source data 6.** PDF containing *Figure 7C* and original scans of the relevant western blot analysis, with cropped areas.

**Source data 7.** Original files for the western blot analysis (*Figure 7D*).

**Source data 8.** PDF containing *Figure 7D* and original scans of the relevant western blot analysis, with cropped areas.

**Source data 9.** Original files for the electrophoretic mobility shift assay (EMSA) analysis (*Figure 7H*).

**Source data 10.** PDF containing *Figure 7H* and original scans of the relevant electrophoretic mobility shift assay (EMSA) analysis, with cropped areas.

**Source data 11.** Original files for the western blot analysis (*Figure 7I*).

**Source data 12.** PDF containing *Figure 7I* and original scans of the relevant western blot analysis, with cropped areas.

**Source data 13.** The ratio between firefly luciferase intensity and Renilla luciferase intensity (*Figure 7J*).

**Figure supplement 1.** $Ca^{2+}$ ionophore A23187 promotes the transformation of fast-to-slow muscle fiber by calcineurin (CaN) signaling pathway.

**Figure supplement 1—source data 1.** Original files for the western blot analysis (*Figure 7—figure supplement 1A*).

**Figure supplement 1—source data 2.** PDF containing *Figure 7—figure supplement 1A* and original scans of the relevant western blot analysis, with cropped areas.

**Figure supplement 1—source data 3.** The mRNA expression of muscle fiber type in C2C12 myotubes (*Figure 7—figure supplement 1B*).

**Figure supplement 1—source data 4.** Original files for the western blot analysis (*Figure 7—figure supplement 1C*).

**Figure supplement 1—source data 5.** PDF containing *Figure 7—figure supplement 1C* and original scans of the relevant western blot analysis, with cropped areas.

diet conditions may promote fat thermogenesis and browning to facilitate fat breakdown. Interestingly, it was found that capsaicin alleviated obesity and promoted white fat browning by activating the TRPV1 (*Baskaran et al., 2016*). Our study found that eugenol promoted TRPV1 mRNA expression in adipose tissue, indicating that eugenol may promote white fat browning through TRPV1 activation.

Exercise mimetics may exert beneficial effects on the body by promoting the release of myokines (*Fan and Evans, 2017*). Some myokines, such as IL-13 (*Knudsen et al., 2020*), IL-15 (*Quinn et al., 2013*), and neurturin (*Correia et al., 2021*), have been well documented to improve metabolic homeostasis, promote fast-to-slow muscle fiber transformation, and improve endurance capacity. However, the regulatory mechanisms for myokines expression are largely unclear. Since myokines are usually promoted by muscle contraction, and $Ca^{2+}$ is the main signal in muscle contraction, CaN, downstream of $Ca^{2+}$, has been considered to regulate myokine expression. CaN has been reported to regulate the expression of myokine IL-6 (*Banzet et al., 2005*; *Banzet et al., 2007*). Additionally, the transcriptional activation activity of myokines IL-4 and IL-13 is thought to be promoted by transcription factor NFATc2, which is downstream of CaN (*Jacquemin et al., 2007*; *Zádor, 2008*). Based on the correlation and TFBS analysis, it was found that IL-15 is most likely to be regulated by the CaN/NFATc1 signaling pathway in our study. IL-15 has been reported as a myokine that improves fatty acid

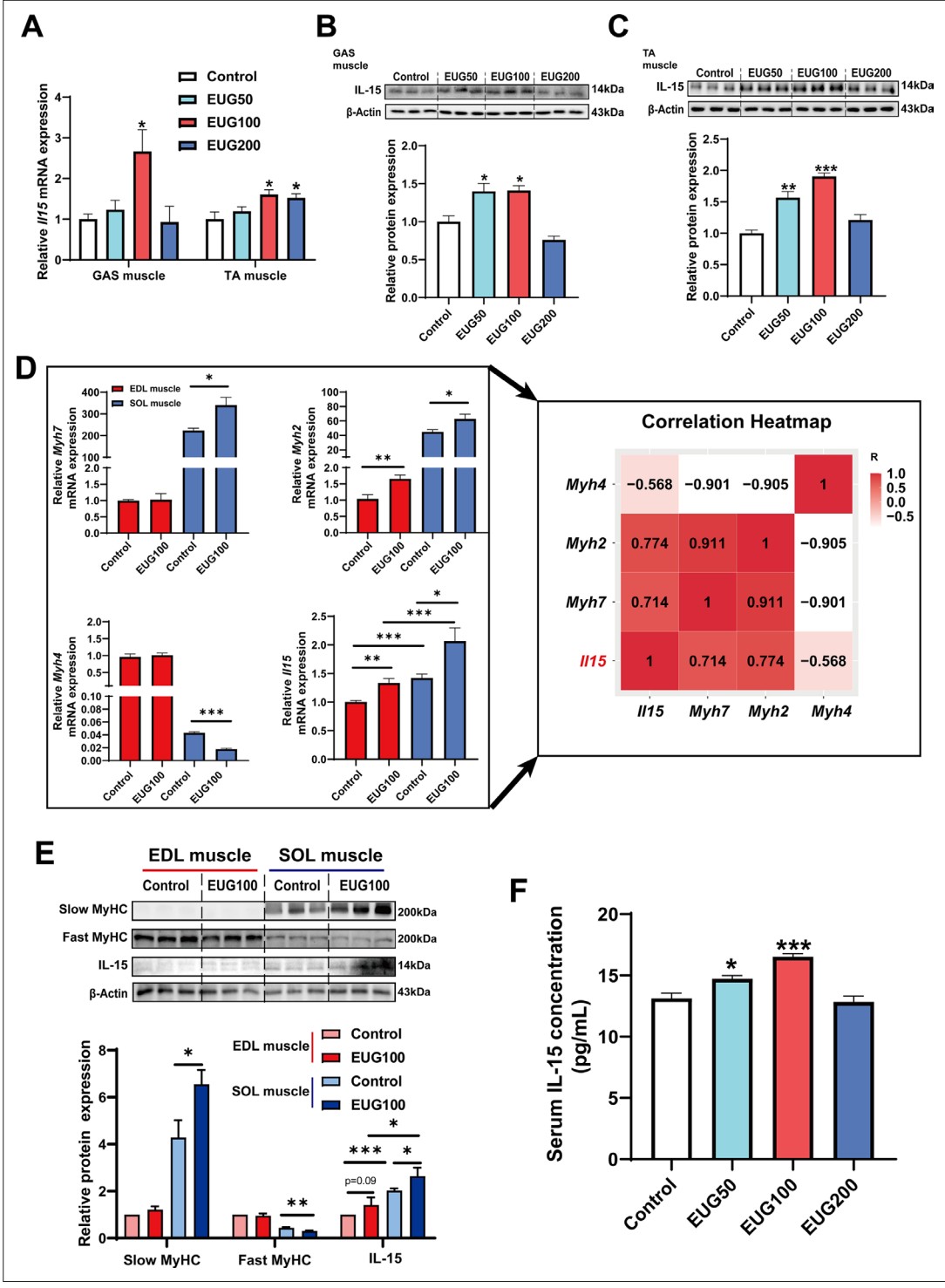

**Figure 8.** Eugenol promotes the expression and secretion of IL-15 in skeletal muscle of mice. (**A**) The *IL-15* mRNA expression in gastrocnemius (GAS) and tibialis anterior (TA) muscle. (**B, C**) The IL-15 protein expression in GAS and TA muscle. (**D**) Left: The mRNA expression of *Myh7*, *Myh2*, *Myh4,* and *Il15* in extensor digitorum longus (EDL) and soleus (SOL) muscle of mice. Right: Correlation analysis of gene expression values performed by linear regression with Pearson's correlation coefficient (r). Color gradient represents correlation coefficient with darker colors indicating higher positive correlation. (**E**) The protein expression of slow myosin heavy chain (MyHC), fast MyHC, and IL-15. (**F**) The concentration of IL-15 in serum. For A and D, N=6 per group. For B and C and E, N=3 per group. For F, N=8 per group. One-way ANOVA test was used to determine statistical significance. *p<0.05, **p<0.01, and ***p<0.001.

*Figure 8 continued on next page*

*Figure 8 continued*

The online version of this article includes the following source data for figure 8:

**Source data 1.** The *IL-15* mRNA expression in gastrocnemius (GAS) and tibialis anterior (TA) muscle (*Figure 8A*).

**Source data 2.** Original files for the western blot analysis (*Figure 8B*).

**Source data 3.** PDF containing *Figure 8B* and original scans of the relevant western blot analysis, with cropped areas.

**Source data 4.** Original files for the western blot analysis (*Figure 8C*).

**Source data 5.** PDF containing *Figure 8C* and original scans of the relevant western blot analysis, with cropped areas.

**Source data 6.** The mRNA expression and correlation analysis of gene expression values (*Figure 8D*).

**Source data 7.** Original files for the western blot analysis (*Figure 8E*).

**Source data 8.** PDF containing *Figure 8E* and original scans of the relevant western blot analysis, with cropped areas.

**Source data 9.** The concentration of IL-15 in serum (*Figure 8F*).

utilization, insulin sensitivity, and endurance capacity, and prevents obesity and diabetes (*Nadeau and Aguer, 2019*). In different cell models, CsA-mediated inhibition of CaN decreased the release of IL-15 (*Cho et al., 2002*; *Cho et al., 2007*). Furthermore, an increase in CaN expression in 3T3-L1 cells was accompanied by an increase in IL-15 expression (*Almendro et al., 2009*). Our study found that the CaN/NFATc1 signaling pathway contributes to the eugenol-promoted expression of IL-15. Furthermore, we observed that the expression of IL-15 was higher in the slow-twitch SOL muscle than in the fast-twitch EDL muscle, suggesting that an increase in muscle oxidative phenotype may further promote the release of IL-15. However, the mechanism underlying how IL-15 promotes the muscle oxidative phenotype remains unclear and requires further investigation.

Apparently, an interesting finding throughout our in vitro and in vivo study was that the high doses of eugenol (200 mg/kg for mice and 200 µM eugenol for C2C12 myotubes) had no effect on TRPV1-mediated CaN/NFATc1 signaling pathway, IL-15 expression, and muscle fiber type. We suspected that high doses of eugenol may cause desensitization of TRPV1 in skeletal muscle under our study conditions. This indirectly verified that the activation of TRPV1 does promote IL-15 expression and fast-to-slow muscle fiber transformation type. It is well known that continuous activation of TRPV1 by capsaicin causes TRPV1 desensitization, which is considered to underlie the analgesic effects of capsaicin (*Koplas et al., 1997*). As previously reported, TRPV1 agonists promoted TRPV1 desensitization in a dose- and time-dependent manner by targeting lysosomes to degrade TRPV1 in HEK293 cells (*Sanz-Salvador et al., 2012*). A hypothetical model suggested that upon TRPV1 opening to allow $Ca^{2+}$ influx, $Ca^{2+}$/CaM binds to the CaM-binding domain of TRPV1, leading to channel inactivation, and CaN dephosphorylates TRPV1 to desensitize TRPV1 (*Hasan and Zhang, 2018*). However, most results on TRPV1 desensitization were obtained by capsaicin treatment of cells, and it is unknown whether high-dose eugenol promotes TRPV1 desensitization through a similar mechanism. Therefore, it is essential to select the appropriate dose of eugenol for the application of TRPV1 activators.

In conclusion, our findings indicate that eugenol can promote the transformation of fast-to-slow muscle fibers and induce the myokine IL-15 expression through the TRPV1-mediated CaN/NFATc1 signaling pathway. Moreover, our study is the first to demonstrate that the expression of IL-15 is regulated by the CaN/NFATc1 signaling pathway. These results suggest that eugenol may have potential as a novel exercise mimetic and that TRPV1 may represent a promising therapeutic target for metabolic disorders.

# Materials and methods
## Animals, treatments, and sample collection

A total of eighty 4-week-old male C57BL/6J mice (Dashuo Experimental Animal Co. Ltd., Chengdu, China) were divided into four treatments (n=20) using a simple randomization method. The control group were fed a basal diet supplemented with 0% eugenol (EUG, purity ≥98%, Sigma, St. Louis, MO, USA), other groups were fed a basal diet supplemented with 50, 100, and 200 mg/kg eugenol,

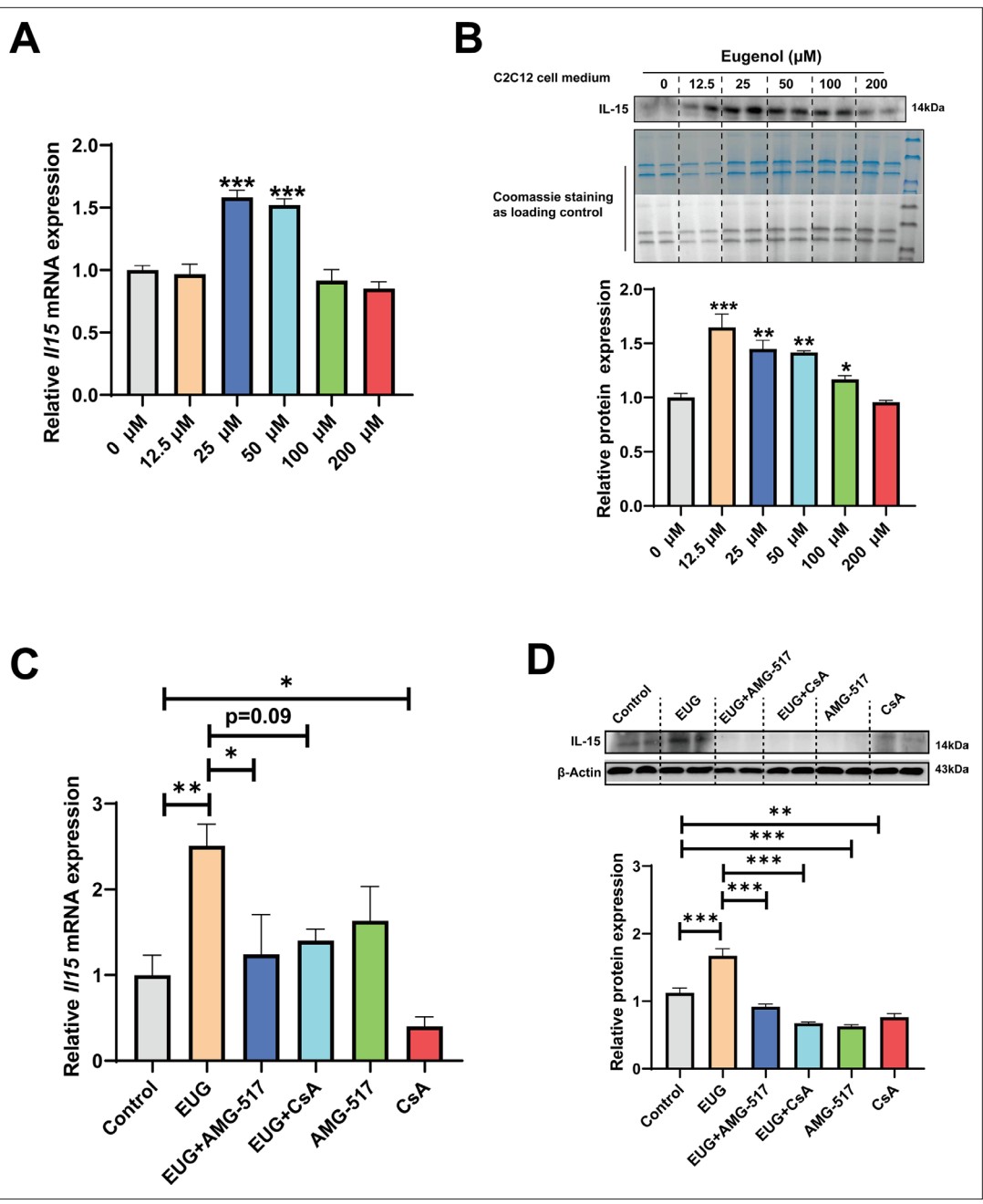

**Figure 9.** Eugenol promotes IL-15 level through TRPV1-mediated CaN/NFATc1 signaling pathway. C2C12 myotubes were treated by 0–200 eugenol for 1 day after 4 days of differentiation. (**A**) The effect of eugenol on *Il15* mRNA expression in C2C12 myotubes. (**B**) The effect of eugenol on IL-15 protein expression in the C2C12 cell medium. Coomassie staining as loading control. (**C, D**) C2C12 myotubes were treated by 25 μM eugenol and 1 μM TRPV1 inhibitor AMG-517 or 0.5 μM calcineurin (CaN) inhibitor cyclosporine A (CsA) for 1 day after 4 days of differentiation. The mRNA and protein expression of IL-15. N=4 per group. One-way ANOVA test was used to determine statistical significance. *p<0.05, **p<0.01, and ***p<0.001.

The online version of this article includes the following source data and figure supplement(s) for figure 9:

**Source data 1.** *IL-15* mRNA expression in C2C12 myotubes (***Figure 9A***).

**Source data 2.** Original files for the western blot analysis (***Figure 9B***).

**Source data 3.** PDF containing ***Figure 9B*** and original scans of the relevant western blot analysis, with cropped areas.

**Source data 4.** *IL-15* mRNA expression in C2C12 myotubes (***Figure 9C***).

*Figure 9 continued on next page*

*Figure 9 continued*

**Source data 5.** Original files for the western blot analysis (*Figure 9D*).

**Source data 6.** PDF containing *Figure 9D* and original scans of the relevant western blot analysis, with cropped areas.

**Figure supplement 1.** Representative immunofluorescence images of IL-15.

respectively (EUG50, EUG100, and EUG200). All mice were housed in individual cages (23°C ± 2°C, 12 hr light/12 hr dark cycle) and provided with free access to feed and water. The experiment lasted for 4 weeks. The weight of mice was measured every week. At the end of the experiment, 14 mice (n=14) anesthetized with $CO_2$ were sacrificed. After being weighed and photographed, skeletal muscle (including TA, GAS, SOL, and EDL), fat (including iWAT, gWAT, BAT) were collected and stored at –80°C for subsequent analyses. All procedures of animal experiments were performed according to the protocols approved by the Animal Care Advisory Committee of Sichuan Agricultural University under permit No. YYS20200929.

## Exhausting swimming test

The forced swimming capacity test was employed in this study to evaluate the effects of eugenol on endurance capacity in mice. A total of thirty 4-week-old mice were divided into the control and EUG100 group (n=15). After 4 weeks of feeding, the mice with a load of lead wire (7% of body weight) attached to its tail were placed in individual swimming pools (25°C ± 1°C, 35 cm depth). Exhaustive swimming time was immediately recorded when the mice failed to return to the surface continuously over a 7 s time frame and showed a lack of coordinated movements.

## Cell lines

The C2C12 cell lines, sourced from the Shanghai Cell Bank of the Chinese Academy of Sciences, have been authenticated using the STR profiling method and tested negative for mycoplasma.

## Cell culture and treatments

C2C12 cells (Shanghai Cell Bank, Chinese Academy of Sciences, passages 3–8) were cultured in Dulbecco's modified Eagle medium (Invitrogen, Carlsbad, CA, USA) supplemented with 10% fetal bovine serum (Gibco, Paisley, Scotland, UK), 100 mg/L streptomycin and 100 U/mL penicillin (Gibco) at 37°C in a 5% $CO_2$ atmosphere. When cells reached ~80% confluence, 2% horse serum (Gibco) replaced 10% fetal bovine serum to induce differentiation. For eugenol treatment, cells were treated with 0, 12.5, 25, 50, 100, 200 μM EUG for 1 day after 4 days of differentiation. For $Ca^{2+}$ ionophore A23187 (Sigma) treatment, cells were treated with 0, 0.1, 0.5, 1 μM A23187 for 16 hr after 2 days of differentiation, or cells were treated with 0.5 μM A23187 and 0.5 μM CsA. For the following mechanism studies, cells were treated with 25 μM EUG and 1 μM TRPV1 inhibitor AMG-517 or 0.5 μM CaN inhibitor cyclosporin A (CsA, Sigma).

## Cell viability assay

Cell viability was analyzed using Cell-Counting Kit-8 (Beyotime, Jiangsu, China) to determine the safe dose of EUG on C2C12 myotubes. Briefly, cells were treated with EUG (0, 12.5, 25, 50, 100, 200, 400, 800, 1600 μM) for 1 day after 4 days of differentiation, 10 μL CCK-8 solution was then added to each well and then incubated for 1 hr at 37°C. After incubation, the OD value was immediately measured at 450 nm using the SpectraMax 190 Absorbance Plate Reader.

## Measurement of intracellular calcium ion ($Ca^{2+}$)

After 1 day of treatment with C2C12 myotubes in 24-well cell culture plates, the cells were digested using Trypsin-EDTA solution and transferred from each well into a centrifuge tube. The Trypsin-EDTA solution was removed and the cells were washed three times with a calcium-free PBS solution. Following this, the PBS solution was removed and 200 μL of 5 μM $Ca^{2+}$ fluorescent probe Fluo-4 (Beyotime) was added to the cells and incubated at 37°C for 30 min. After the incubation, the probe was removed by washing the cells three times with PBS. Finally, $Ca^{2+}$ fluorescence was detected using

flow cytometry (FACSVerse, BD Biosciences, East Rutherford, NJ, USA) and analyzed using FlowJo 10.0.7 software.

## Gene expression and mitochondrial DNA qPCR

Total RNA was extracted using RNA Isolater Total RNA Extraction Reagent (Vazyme, Nanjing, China) and genomic DNA was extracted using mammalian genomic DNA extraction kit (Beyotime) according to the instructions. After measuring the concentration of total RNA, total RNA reverse-transcribed to cDNA using HiScript II Q RT Supermix (Vazyme). qPCR was performed using ChamQ SYBR Color qPCR Master Mix (Vazyme) on a 7900 HT Real-time PCR system (384-cell standard block) (Applied Biosystems). Relative mtDNA was quantified by qPCR using primers for mitochondrially encoded *Nd1* normalized to nuclear-encoded *36b4 (Rplp0)* DNA. And *Gapdh* was used as an endogenous control for normal qPCR. The primer sequences are listed in *Supplementary file 1*.

## Protein extraction and western blot

RIPA lysis buffer (Beyotime) was used to extract total protein. The nuclear protein was extracted using NE-PER Nuclear and Cytoplasmic Extraction Reagents (Thermo Fisher). Protein from conditioned media was extracted using methanol-chloroform precipitation method as previously described (*Jakobs et al., 2013*). Protein concentration was determined by the BCA assays, and then proteins were transferred to a nitrocellulose membrane using a wet Trans-Blot system (Bio-Rad, Hercules, CA, USA). The primary antibodies used were anti-slow MyHC (Sigma, cat. no. M8421), anti-fast MyHC (Sigma, cat. no. M4276), anti-TRPV1 (Alomone, cat. no. ACC030), anti-Calcineurin A (CnA, Abcam, cat. no. ab90540), anti-NFATc1 (Cell Signaling Technology, cat. no. #8032), anti-IL-15 (R&D Systems, cat. no. AF447), anti-PGC-1α (Proteintech, cat. no. 66369-1-Ig), anti-NDUFA9 (GeneTex, cat. no. GTX132978), anti-SDHA (GeneTex, cat. no. GTX636098), anti-UQCRC1 (GeneTex, cat. no. GTX630393), anti-MTCO1(Bioss, cat. no. bs-3953R), anti-ATP5B (GeneTex, cat. no. GTX132925), anti-FABP1 (Cell Signaling Technology, cat. no. #13368), anti-UCP-1 (Proteintech, cat. no. 23673-1-AP), anti-PRDM16 (R&D Systems, cat. no. AF6295), anti-β-actin (TransGen, cat. no. HC201-01), and anti-Histone H3 (Beyotime, cat. no. AF0009). Coomassie staining is depicted as loading control for conditioned media protein (*Welinder and Ekblad, 2011*). The signal was visualized using a Clarity Western ECL Substrate (Bio-Rad, Hercules, CA, USA) and a ChemiDoc XRS Imager System (Bio-Rad). The target band density was identified using Image Lab 5.1 (Bio-Rad).

## Immunofluorescence

After treatment, C2C12 myotubes were washed three times (5 min each time) with phosphate-buffered saline (PBS) and fixed in Immunol Staining Fix Solution (Beyotime) for 20 min. Then C2C12 myotubes were permeabilized with 0.5% Triton X-100 for 20 min, blocked with blocking buffer for 2 hr at 37°C, incubated with the primary antibodies including slow MyHC (1:50, Sigma, Cat. No. M8421) for 16 hr, and incubated with the fluorescent secondary antibody (1:1000, Cell Signaling, USA) for 2 hr at 37°C. Finally, 4,6-diamidino-2-phenylindole (Beyotime) was used to stain cell nucleus for 10 min at room temperature. A positive signal was detected and captured using fluorescence microscopy (Lecia DMI4000 B).

## Molecular-protein docking and DNA-protein docking

The protein structures of TRPV1 (PDB codes: 5IS0) and NFATc1 (PDB codes: 1A66) were downloaded from the Research Collaboratory for Structural Bioinformatics Protein Data Bank database (RCSB PDB, https://www.rcsb.org/) (*Burley et al., 2019*). The structure of eugenol (ZINC1411) was downloaded from the ZINC database (https://zinc.docking.org/substances/). The DNA structure was generated using Discovery Studio 2019 software (Discovery Studio 2019; BIOVIA; San Diego, CA, USA). Discovery Studio 2019 software was used to perform molecular-protein docking. HDOCK (http://hdock.phys.hust.edu.cn/) was used to perform DNA-protein docking.

## Prediction of TFBS

The gene promoter sequences of mouse (upstream 2 kb) were acquired from National Center for Biotechnology Information. hIFtarget (https://guolab.wchscu.cn/hTFtarget/#!/) and JASPAR (https://

jaspar.elixir.no/) applied to predict the potential NFATc1 binding sites at the promoter of genes. A putative binding sites predicted by both the tools were selected for further EMSA probes design.

## Electrophoretic mobility shift assay

After synthesizing the single-stranded probes for EMSA, annealing buffer for DNA oligos (5×) (Beyotime) was used to anneal to form double-stranded DNA probes. EMSA were performed using chemiluminescent EMSA Kit (Beyotime) according to the instructions. Briefly, 10 µL reaction system with nuclease-free water, EMSA/Gel-Shift Binding Buffer, nuclear protein, and probe was transferred to a nitrocellulose membrane using a wet Trans-Blot system. The reaction system includes negative control reaction (no nuclear protein), sample reaction, probe cold competition reaction (100-fold unlabeled probe), mutation probe cold competition reaction (100-fold unlabeled mutation probe). The signal was visualized using a Clarity Western ECL Substrate (Bio-Rad, Hercules, CA, USA) and a ChemiDoc XRS Imager System (Bio-Rad).

## Plasmid construction and extraction

The plasmid of pcDNA3.1 vector, pCDNA3.1-NFATc1 (the NFATc1 coding sequences were inserted into the pcDNA3.1 vector), pGL3 vector, and pGL3-IL15 (the sequence of 2 kb upstream of the IL-15 promoter sequence were inserted into the pGL3 vector) were from Tsingke Biotechnology Co., Ltd (Beijing, China). Plasmid amplification is provided in *Escherichia coli* bacteria cells. Plasmid Maxi Preparation Kit for All Purpose (Beyotime) was used to extract plasmids.

## Dual-luciferase reporter gene assay

HEK293T cells in 48-well plates were transfected with pcDNA3.1 together with pGL3 or pcDNA3.1-NFATc1 together with pGL3 or pcDNA3.1 together with pGL3-IL15 or pcDNA3.1-NFATC1 together with pGL3-IL15. Luciferase activity was detected by a Dual-Luciferase assay kit (Beyotime) with Glomax 96 microplate luminometer (Promega) in luminometer mode. The raw values of firefly luciferase were normalized to Renilla luciferase.

## Enzyme activities analysis

The tissue and C2C12 myotube were homogenized in saline and centrifuged at $3500 \times g$ 4°C for 10 min. The supernatant was carefully transferred to a centrifuge tube. Total protein concentration was determined by the BCA assays. The enzyme activity of LDH, MDH, SDH, and CaN was measured using commercial assay kits (Nanjing Jiancheng Bioengineering Institute, Nanjing, China).

## Statistical analyses

SAS 9.4 software was used to perform one-way ANOVA and t test. After the normality and homogeneity test, one-way ANOVA followed by Duncan's multiple-range test was performed for multiple-groups comparisons. Student's t test was performed for two-groups comparisons. Correlation analysis was performed by Pearson's correlation coefficient analysis. Statistical methods were not used to predetermine sample size. GraphPad Prism 8.0 (GraphPad Software, Inc, San Diego, CA, USA) software was used to draw column charts and Omicstudio tools (https://www.omicstudio.cn/tool.) was used to draw heatmaps. Data were expressed as mean ± SEM. Statistical significance was defined as $^{\#}p < 0.1$, $^{*}p < 0.05$, $^{**}p < 0.01$, and $^{***}p < 0.001$ for all figures.

## Acknowledgements

This work was supported by the National Key R&D Program of China (No. 2023YFD1301302), the National Natural Science Foundation of China (No. 32372901), the Natural Science Foundation of Sichuan Province (No. 2023NSFSC0238), and the Sichuan Science and Technology Program (No. 2021ZDZX0009).

## Additional information

### Funding

| Funder | Grant reference number | Author |
|---|---|---|
| National Key Research and Development Program of China | 2023YFD1301302 | Zhiqing Huang |
| National Natural Science Foundation of China | 32372901 | Zhiqing Huang |
| Natural Science Foundation of Sichuan Province | 2023NSFSC0238 | Zhiqing Huang |
| Sichuan Science and Technology Program | 2021ZDZX0009 | Zhiqing Huang |

The funders had no role in study design, data collection and interpretation, or the decision to submit the work for publication.

### Author contributions

Tengteng Huang, Data curation, Formal analysis, Investigation, Writing - original draft; Xiaoling Chen, Conceptualization, Supervision, Methodology, Project administration; Jun He, Ping Zheng, Yuheng Luo, Aimin Wu, Hui Yan, Bing Yu, Daiwen Chen, Methodology; Zhiqing Huang, Conceptualization, Supervision, Funding acquisition, Methodology, Writing - review and editing

### Author ORCIDs

Tengteng Huang ⓘ http://orcid.org/0000-0003-3583-1835
Hui Yan ⓘ http://orcid.org/0000-0001-7476-3914
Zhiqing Huang ⓘ http://orcid.org/0000-0001-5092-9297

### Ethics

All procedures of animal experiments were performed according to the protocols approved by the Animal Care Advisory Committee of Sichuan Agricultural University under permit No. YYS20200929.

Reviewer #1 (Public Review): https://doi.org/10.7554/eLife.90724.3.sa1
Reviewer #2 (Public Review): https://doi.org/10.7554/eLife.90724.3.sa2
Author response https://doi.org/10.7554/eLife.90724.3.sa3

---

## Additional files

### Supplementary files
• MDAR checklist

• Supplementary file 1. The primer sequences used in the study.

### Data availability

All data generated or analyzed during this study are included in the manuscript and supporting files.

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
