## [Editor Report · eLife assessment]

This **useful** paper addresses a novel exercise mimetic agent on muscle exercise and performance. While the data provided are interesting, the evidence is **incomplete**, as much of it is correlative.

---

## [Referee Report · Reviewer #1 (Public Review)]

Summary:

In this manuscript, Huang et al have investigated the exercise mimetic role of Eugenol (a natural product) in skeletal muscle and whole-body fitness. The authors report that Eugenol facilitates skeletal muscle remodeling to a slower/oxidative phenotype typically associated with endurance. Eugenol also remodels the fat driving browning the WAT. In both skeletal muscle and fat Eugenol promotes oxidative capacity and mitochondrial biogenesis markers. Eugenol also improves exercise tolerance in a swimming test. Through a series of in vitro studies the authors demonstrate that eugenol may function through the trpv1 channel, Ca mobilization, and activation of CaN/NFAT signaling in the skeletal muscle to regulate slow-twitch phenotype. In addition, Eugenol also induces several myokines but mainly IL-15 through which it may exert its exercise mimetic effects. Overall, the manuscript is well-written, but there are several mechanistic gaps, physiological characterization is limited, and some data are mostly co-relative without vigorous testing (e.g. link between Eugenol, IL15 induction, and endurance). Specific major concerns are listed below.

Strengths:

A natural product activator of the TRPV1 channel that could elicit exercise-like effects through skeletal muscle remodeling. Potential for discovering other mechanisms of action of Eugenol.

Weaknesses:

(1) Figure 1: Histomorphological analysis using immunostaining for type I, IIA, IIX, and IIB should be performed and quantified across different muscle groups and also in the soleus. Fiber type switch measured based on qPCR and Westerns does not sufficiently indicate the extent of fiber type switch. Better images for Fig. 1c should be provided.

(2) Figure 2: Histomorphological analysis for SDH and NADH-TR should be performed and quantified in different muscle groups. Seahorse or oroborous respirometry experiments should be performed to determine the actually increase in mitochondrial respiratory capacity either in isolated mitochondria or single fibers from vehicle and Eugenol-treated mice. Em for mitochondrial should be added to determine the extent of mitochondrial remodeling. The current data is insufficient to indicate the extent of mitochondrial or oxidative remodeling.

(3) Figure 2: Gene expression analysis is limited to a few transcriptional factors. A thorough analysis of gene expression through RNA-seq should be performed to get an unbiased effect of Eugenol on muscle transcriptome. This is especially important because eugenol is proposed to work through CaN/NFAT signaling, major transcriptional regulators of muscle phenotype.

(4) I suggest the inclusion of additional exercise or performance testing including treadmill running, wheel running, and tensiometry. Quantification with a swimming test and measurement of the exact intensity of exercise, etc. is limited.

(5) In addition to muscle performance, whole-body metabolic/energy homeostatic effects should also be measured to determine a potential increase in aerobic metabolism over anaerobic metabolism.

(6) For the swimming test and other measurements, only 4 weeks of vehicle vs. Eugenol treatment was used. For this type of pharmacological study, a time course should be performed to determine the saturation point of the effect. Does exercise tolerance progressively increase with time?

(7) The authors should also consider measuring adaptation to exercise training with or without Eugenol.

(8) Histomorphological analysis of Wat is also lacking. EchoMRI would give a better picture of lean and fat mass.

(9) The experiments performed to demonstrate that Eugenol functions through trpv1 are mostly correlational. Some experiments are needed with trpv1 KO or KD instead of inhibitor. Similarly, KD for other trpv channels should be tested (at least 1-4 that seem to be expressed in the muscle). Triple KO or trpv null cells should be considered to demonstrate that eugenol does not have another biological target.

(10) Eugenol + trpv1 inhibition studies are performed in c2c12 cells and only looks at myofiber genes expression. This is incomplete. Some studies in mitochondrial and oxsphos genes should be done.

(11) The experiments linking Eugenol to ca handling, and calcineurin/nfat activation are all performed in c2c12 cells. There seems to be a link between Eugenol activation and CaN/NFAT activation and fiber type regulation in cells, however, this needs to be tested in mouse studies at the functional level using some of the parameters measured in aims 1 and 2.

(12) The myokine studies are incomplete. The authors show a link between Eugenol treatment and myokines/IL-15 induction. However, this is purely co-relational, without any experiments performed to show whether IL-15 mediates any of the effects of eugenol in mice.

(13) An additional major concern is that it cannot be ruled out that Engenol is uniquely mediating its effects through trpv1. Ideally, muscle-specific trpv1 mice should be used to perform some experiments with Eugenol to confirm that this ion channel is involved in the physiological effects of eugenol.

Comments on revised version:

Unfortunately, in the revision the authors have not addressed any of my comments with new experimental data. For example, some of the histological experiments I suggested are quite easy to do or standardize. Other in vitro experiments could also be conducted to show direct mechanistic link. The current revision does not further improve the manuscript from the 1st submission.

---

## [Referee Report · Reviewer #2 (Public Review)]

Summary:

The authors examined the hypothesis that eugenol promotes myokines release and f skeletal muscles remoulding by activating the TRPV1-Ca2+-calcineurin-NFATc1 signalling pathway. They first showed that eugenol promotes skeletal muscle transformation and metabolic functions in adipose tissues by analysing changes in the expression of mRNA and proteins of relevant representative protein markers. With similar methodologies, they further found that eugenol increases the expression of mRNA and/or proteins of TRPV1, CaN, NFATC1 and IL-15 in muscle tissues. These processes were, however, prevented by inhibiting TRPV1 and CaN. Similar expression changes were also triggered by increasing intracellular Ca2+ with A23187, suggesting a Ca2+-dependent process.

Strengths:

Different proteins markers were used as a readout of the functions of muscles and adipose tissues and mitochondria and analysed at both mRNA and protein levels. The results were mostly consistent. Although the signaling pathway of TRPV1-Ca2+-CaN-NFAT is not new and well documented, they identified IL-15 as a new downstream target of this pathway combined with use of TRPV1 and CaN inhibitors.

Weaknesses:

Most of the evidence is limited to the molecular level lacking direct functional assays and system analysis. It will be interesting to examine the effect of eugenol on metabolic rate in animals and the role of TRPV1 in this process, as eugenol enhanced food intake without effect on body weight. TRPV1 and CaN inhibition prevented IL-15 expression in C2C12 cells (Fig.9). It remain unknown whether the effect is reproducible in native muscle tissues.

It is also unknown how eugenol enhances TRPV1/CaN expression and alters the expression of many other protein markers in muscle and adipose tissues. Are these effects mediated by activated NFAT or by released IL-15 forming a positive feedback loop? It should at least be discussed.

Many protein blots were presented but no molecular weight markers were shown. It is thus difficult to convince others that the protein bands are the right anticipated positions.

---

## [Author Response]

The following is the authors’ response to the original reviews.

**Reviewer #1 (Public Review):**
Weaknesses:(1) Figure 1: Histomorphological analysis using immunostaining for type I, IIA, IIX, and IIB should be performed and quantified across different muscle groups and also in the soleus. Fiber type switch measured based on qPCR and Westerns does not sufficiently indicate the extent of fiber type switch. Better images for Fig. 1c should be provided.

Thanks for your suggestion. In fact, we attempted immunofluorescent staining for Slow MyHC and Fast MyHC in GAS muscle. However, for the majority of our results, we only observed positive expression of Slow MyHC in a small portion of the muscle sections (as shown in the figure below), so we did not present this result.

In addition, due to the size limitations on uploading image files to Biorxiv, we had to compress the images, resulting in lower resolution pictures. We have attempted to submit clearer images in Fig. 1C

**Author response image 1. sa3fig1:** Green: Slow MyHC; Red: Fast MyHC.

(2) Figure 2: Histomorphological analysis for SDH and NADH-TR should be performed and quantified in different muscle groups. Seahorse or oroborous respirometry experiments should be performed to determine the actually increase in mitochondrial respiratory capacity either in isolated mitochondria or single fibers from vehicle and Eugenol-treated mice. Em for mitochondrial should be added to determine the extent of mitochondrial remodeling. The current data is insufficient to indicate the extent of mitochondrial or oxidative remodeling.

That's a good suggestion. However, we regret to inform you that we are unable to present these results due to a lack of relevant experimental equipment and samples.

(3) Figure 2: Gene expression analysis is limited to a few transcriptional factors. A thorough analysis of gene expression through RNA-seq should be performed to get an unbiased effect of Eugenol on muscle transcriptome. This is especially important because eugenol is proposed to work through CaN/NFAT signaling, major transcriptional regulators of muscle phenotype.

Thanks for your suggestion. Indeed, we believe that in terms of reliability and accuracy, RNA-seq is not as good as RT-qPCR. The advantage of RNA-seq lies in its high throughput, making it suitable for screening unknown transcription factor regulatory mechanisms. In this study, the signaling pathways regulating myokines and muscle fiber type transformation are known and limited, with only the CaN/NFATc1 and the AMPK pathway. Since eugenol mainly acts through the Ca2+ pathway, we primarily focus on the CaN/NFATc1 signaling pathway.

(4) I suggest the inclusion of additional exercise or performance testing including treadmill running, wheel running, and tensiometry. Quantification with a swimming test and measurement of the exact intensity of exercise, etc. is limited.

That's a good suggestion. We apologize for being unable to detect this indicator due to a lack of relevant experimental equipment.

(5) In addition to muscle performance, whole-body metabolic/energy homeostatic effects should also be measured to determine a potential increase in aerobic metabolism over anaerobic metabolism.

That's a good suggestion. We apologize for being unable to detect this indicator due to a lack of relevant experimental equipment.

(6) For the swimming test and other measurements, only 4 weeks of vehicle vs. Eugenol treatment was used. For this type of pharmacological study, a time course should be performed to determine the saturation point of the effect. Does exercise tolerance progressively increase with time?

Thanks for your suggestion. Due to the potential damage that exhaustive swimming tests inflict on mice, the tested mice are subsequently eliminated to avoid potential interference with the experiment. Therefore, this experiment is only suitable for conducting tests at individual time points.

(7) The authors should also consider measuring adaptation to exercise training with or without Eugenol.

Thanks for your suggestion. The purpose of this study is to investigate whether eugenol mimics exercise under standard dietary conditions. In our future research, we will consider exploring the effects of eugenol under HFD and exercise conditions.

(8) Histomorphological analysis of Wat is also lacking. EchoMRI would give a better picture of lean and fat mass.

That's a good suggestion. However, we did not collect the slices of WAT tissue, so we are unable to supplement this result, we feel sorry for it. In addition, we apologize for being unable to detect lean and fat mass due to a lack of EchoMRI equipment.

(9) The experiments performed to demonstrate that Eugenol functions through trpv1 are mostly correlational. Some experiments are needed with trpv1 KO or KD instead of inhibitor. Similarly, KD for other trpv channels should be tested (at least 1-4 that seem to be expressed in the muscle). Triple KO or trpv null cells should be considered to demonstrate that eugenol does not have another biological target.

Thanks for your professional suggestion. AMG-517 is a specific inhibitor of TRPV1, with a much greater inhibitory effect on TRPV1 compared to other TRP channels. AMG-517 inhibits capsaicin (500 nM), acid (pH 5.0), or heat (45°C) induced Ca2+ influx in cells expressing human TRPV1, with IC50 values of 0.76 nM, 0.62 nM, and 1.3 nM, respectively. However, the IC50 values of AMG-517 for recombinant TRPV2, TRPV3, TRPV4, TRPA1, and TRPM8 cells are >20 μM (Gavva, 2008). Therefore, we believe that using AMG-517 instead of TRPV1 KO cells is sufficient to demonstrate the involvement of TRPV1 in the function of eugenol.

While this study did not exclude the possibility of other TRP channels' involvement, it was based on the fact that eugenol does not promote mRNA expression of other TRP channels, as shown in Fig4A-C. Indeed, as far as we know, besides TRPV1, the effects of other TRP channels on myofiber type transformation remain unknown. This is an aspect that we plan to investigate in the future.

Reference

Gavva NR, Treanor JJ, Garami A, et al. Pharmacological blockade of the vanilloid receptor TRPV1 elicits marked hyperthermia in humans. Pain. 2008;136(1-2):202-210.

(10) Eugenol + trpv1 inhibition studies are performed in c2c12 cells and only looks at myofiber genes expression. This is incomplete. Some studies in mitochondrial and oxsphos genes should be done.

Thanks for your suggestion. In the inhibition experiment, we additionally examined the expression of mitochondrial complex proteins as shown in Figure 5C. And the relevant description has been added in lines 178-183 and 764-765.

(11) The experiments linking Eugenol to ca handling, and calcineurin/nfat activation are all performed in c2c12 cells. There seems to be a link between Eugenol activation and CaN/NFAT activation and fiber type regulation in cells, however, this needs to be tested in mouse studies at the functional level using some of the parameters measured in aims 1 and 2.

Thank you for your professional suggestion. We will attempt to continue these experiments in future studies.

(12) The myokine studies are incomplete. The authors show a link between Eugenol treatment and myokines/IL-15 induction. However, this is purely co-relational, without any experiments performed to show whether IL-15 mediates any of the effects of eugenol in mice.

Indeed, previous studies have adequately demonstrated the regulation of skeletal muscle oxidative metabolism by IL-15. The initial aim of this experiment was to investigate the mechanism by which eugenol promotes IL-15 expression. Through inhibition assays, EMSA, and dual luciferase reporter gene experiments, we have thoroughly demonstrated that eugenol promotes IL-15 expression via the CaN/NFATc1 signaling pathway, thus establishing a novel link between CaN/NFATc1 signaling and the myokine IL-15 expression. In the subsequent experiments, we plan to knock out IL-15 in eugenol-treated C2C12 cells to explore whether IL-15 mediates the effects of eugenol. This will be another aspect of our investigation.

(13) An additional major concern is that it cannot be ruled out that Engenol is uniquely mediating its effects through trpv1. Ideally, muscle-specific trpv1 mice should be used to perform some experiments with Eugenol to confirm that this ion channel is involved in the physiological effects of eugenol.

As you suggested, we agree that muscle-specific TRPV1 mice should be used to conduct some experiments with eugenol. In our mice experiments, due to the lack of validation of skeletal muscle-specific TRPV1 knockout, we indeed cannot rule out that eugenol is uniquely mediating its effects through TRPV1. We acknowledge this as a limitation of our study. However, due to limitations in research funding and time, we are currently unable to supplement these experiments. Nevertheless, we believe that our results from in vitro experiments using a TRPV1 inhibitor (which selectively inhibits TRPV1) provide evidence of eugenol's action through TRPV1.

**Reviewer #2 (Public Review):**
Weaknesses:(1) Apart from Fig.2A and 2B, they mostly utilised protein expression changes as an index of tissue functional changes. Most of the data supporting the conclusions are thus rather indirect. More direct functional evidence would be more compelling. For example, a lipolysis assay could be used to measure the metabolic function of adipocytes after eugenol treatment in Fig.3. Functional activation of NFAT can be demonstrated by examining the nuclear translocation of NFAT.

Thank you for your professional suggestion. Indeed, as shown in Figure 4G-I, we detected the expression of NFATc1 in the nucleus to illustrate its nuclear translocation.

(2) To further demonstrate the role of TRPV1 channels in the effects of eugenol, TRPV1-deficient mice and tissues could also be used. Will the improved swimming test in Fig. 2B and increased CaN, NFAT, and IL-15 triggered by eugenol be all prevented in TRPV1-lacking mice and tissues?

Thank you for your professional suggestion. We agree that muscle-specific TRPV1 mice should be used to conduct some experiments with eugenol. However, due to limitations in research funding and time, we are currently unable to supplement these experiments.

(3) Direct evidence of eugenol activation of TRPV1 channels in skeletal muscles is also lacking. The flow cytometry assay was used to measure Ca2+ changes in the C2C12 cell line in Fig. 5A. But this assay is rather indirect. It would be more convincing to monitor real-time activation of TRPV1 channels in skeletal muscles not in cell lines using Ca2+ imaging or electrophysiology.

Thank you for your professional suggestion. As you suggested, we initially planned to use patch-clamp technique to detect membrane potential changes in skeletal muscle cells under eugenol treatment. However, due to experimental technical limitations, this experiment was not successfully conducted. Therefore, we were compelled to rely solely on flow cytometry to detect Ca2+ levels.

**Reviewer #2 (Recommendations For The Authors):**
(1) Most of the mRNA and protein data are consistent with each other. However, some of them are not obvious. For example, PGC1a mRNA was increased by eugenol in Fig. 2C but not seen in protein in Fig. 2D. Similarly, Complex I and V mRNA was increased in Fig. 2C but not obvious at protein levels in Fig. 2D, even though they claimed that Complex I and V were both upregulated by eugenol (see: line 123). Another example: IL-15 mRNA was increased by EUG100 but not by EUG50 in the GAS muscle in Fig. 8A. However, EUG50 increased IL-15 protein expression in Fig. 8B. Similar conflict was also seen in IL-15 expression in the TA muscle in Fig. 8A and 8C.

Thanks for your question. As shown in the table below, by standardizing with β-Actin, our statistical data indeed indicate that eugenol promotes the expression of Complex I and V proteins (although the upregulation is minimal). Additionally, protein and mRNA expression do not always correlate, which may be due to potential post-transcriptional and post-translational regulation.

**Author response table 1. sa3table1:** 

	Lane1	Lane2	Lane3	Lane4	Lane5	Lane6
beta-Actin	1.000	0.853	0.821	0.859	0.816	0.812
Complex I	1.000	0.739	0.867	0.917	1.034	1.150
Complex V	1.000	0.879	0.856	0.949	0.947	0.985
Complex I/ beta-Actin	1.000	0.867	1.056	1.068	1.268	1.415
Complex V/ beta-Actin	1.000	1.031	1.043	1.105	1.161	1.213

(2) Line 115: Figure 2A should be Figure 2B; Line 119: Figure 2B should be Figure 2A. Alternatively, swap Fig2A with Fig. 2B.

Thanks for your correction, we have revised the relevant content in lines 111-113 and 724-725.

(3) Abbreviations of ADF and ADG in Fig. 3A should be defined.

Thank you for your suggestion. We have defined these abbreviations in lines 123-125.

(4) Line 154: TRPV1 mRNA expression was promoted by 25 and 50uM eugenol, not by 12.5uM.

Thank you for your correction. We have revised it in line 150.

(5) Line 173: Increased expression of NFAT suggests that NFAT is activated. This is a rather weak statement. It is more convincing to show the nuclear translocation of NFAT by eugenol treatment.

Thank you for your correction. We have revised the describtion in line 166.

(6) Line 185: The data showing EUG increased slow MyHC fluorescence intensity in Fig. 5D are not clear at all. Quantification is required.

Thank you for your suggestion. We have attempted to submit clearer images in Figure 5E, and the quantification have been provided.

(7) Line 235: IL-15 expression is positively correlated with MyHC IIa, suggesting IL-15 is a slow muscle myokine (See line 2398). However, MyHC IIa is a marker of fast muscle fibres (see line 50).

Thank you for your correction. As you pointed, MyHC IIa is fast-twitch oxidative muscle fiber. We have replaced ‘slow’ with ‘oxidative’ in line 235.

(8) Fig.9C and 9D show that inhibition of TRPV1 and CaN attenuated the upregulation of IL-15 mRNA and protein by eugenol in C2C12 cell line. This result is important in demonstrating the link of TRPV1 and CaN to IL-15. It will be more interesting and physiologically relevant to perform this experiment in primary skeletal muscle cells isolated from mice.

Thank you for your suggestion. This is indeed an interesting idea. We will attempt to continue our experiments in mice and primary porcine muscle cells in future studies.

(9) It is concerning that 4-week-old male mice were used for the study. The 4-week-old mice are immature. Adult mice over 8 weeks should be used. It is thus unknown whether the findings are broadly applicable to adult age.

Thanks for your professional question. Age indeed has an impact on the muscle fiber type in mammals. Based on previously observed patterns of muscle fiber changes with age in various mammals (Katsumata et al., 2021; Pandorf et al., 2012; Hill et al., 2020), we believe that changes in muscle fiber types occur more frequently in juvenile mammals, mainly manifesting as a sharp increase in fast muscle fibers. Therefore, interventions during the juvenile stage might be more effective in promoting the transformation of fast to slow muscle fibers. As a result, in most of our group's research using nutritional interventions to regulate muscle fiber types, we tend to start interventions from the age of 4 weeks in mice. If we began intervention at 8 weeks, we speculate that the effectiveness would not be as potent as starting at 4 weeks. Below are the patterns of muscle fiber changes with age in various mammalian models, provided for reference:

(1) Changes in muscle fiber types with age in pigs:

As shown in the following figure, there is a dramatic change in the muscle fiber types 12 days post birth in pigs, especially with a sharp increase in fast muscle fibers, which continues until day 45. After 45 days of age, the changes in muscle fiber types become relatively gradual.

**Author response table 2. sa3table2:** Developmental change of proportions of muscle fiber types in Longissimus dorsi muscle determined by histochemical analysis for myosin adenosine triphosphatase activity (%). Least squares means and pooled standard errors (n = 3). MHC, myosin heavy chain; ND, not detected. *p<0.10, **p<0.01 Least square means followed by different letters on the same row are significantly different (p<0.05).

	d 1	d 12	d 26	d 45	d 75	Pooled SE	
MHC 1	7.2	8.2	12.1	14.8	11.2	1.8	*
MHC 2 a	92.8^a^	23.2^b^	18.6^b^	4.1^c^	3.1^c^	4.3	**
MHC 2 x or 2b	ND	68.6^b^	69.3^b^	81.1^ab^	85.7^a^	4.8	**

Reference:

Katsumata, M., Yamaguchi, T., Ishida, A., & Ashihara, A. (2017). Changes in muscle fiber type and expression of mRNA of myosin heavy chain isoforms in porcine muscle during pre- and postnatal development. Animal science journal, 88(2), 364–371.

(2) Changes in muscle fiber types with age in rats:

As illustrated in the subsequent figure, the muscle fiber types in rats undergo significant changes before 20 days of age (3-week-old), notably with a pronounced increase in type IIb fast-twitch fibers. After reaching 20 days of age, the changes in type IIb muscle fibers tend to stabilize and become more gradual.

**Author response image 2. sa3fig2:** 

Reference:

Pandorf, C. E., Jiang, W., Qin, A. X., Bodell, P. W., Baldwin, K. M., & Haddad, F. (2012). Regulation of an antisense RNA with the transition of neonatal to IIb myosin heavy chain during postnatal development and hypothyroidism in rat skeletal muscle. American journal of physiology. 302(7), R854–R867.

(3) Changes in muscle fiber types with age in mice:

As depicted in the following figure, when comparing 10-week-old mice to 78-week-old aged mice, there are no significant changes in muscle fiber types.

**Author response image 3. sa3fig3:** 

Reference:

Hill, C., James, R. S., Cox, V. M., Seebacher, F., & Tallis, J. (2020). Age-related changes in isolated mouse skeletal muscle function are dependent on sex, muscle, and contractility mode. American journal of physiology. Regulatory, integrative and comparative physiology, 319(3), R296–R314.